

# State of the art of fragility analysis for major building types in China with implications for intensity-PGA relationships

Danhua Xin[1,2*], James Edward Daniell[1,3], Friedemann Wenzel[1]

[1]Center for Disaster Management and Risk Reduction Technology (CEDIM) and Geophysical Institute, Karlsruhe Institute of Technology, Hertzstrasse 16, 76187, Karlsruhe, Germany

[2]China Scholarship Council (CSC), Level 13, Building A3, No.9 Chegongzhuang Avenue, 100044, Beijing, P.R. China

[3]General Sir John Monash Scholar, The General Sir John Monash Foundation, Level 5, 30 Collins Street, Melbourne,Victoria, 3000, Australia

*Correspondence to*: Danhua Xin (danhua.xin@kit.edu)

**Abstract**. The evaluation of the seismic fragility of buildings is one key task of earthquake safety and loss assessment. Many research reports and papers have been published over the past four decades that deal with the vulnerability of buildings to ground motion caused by earthquakes in China. We scrutinize 69 papers with studies of building damage for magnitude≥4.7 events occurred in densely populated areas starting with the 1975 M7.5 Haicheng earthquake. They represent observations where macroseismic intensities have been determined according to the Chinese Official Seismic Intensity Scale. From these many studies we derive the most representative fragility functions (dependent on intensity) for 4 damage limit states of two most widely distributed building types: masonry and reinforced concrete. We also inspect 18 papers that provide analytical fragility curves (dependent on PGA) for the same damage classes and building categories. Finally, we check the consistency of fragilities as functions of intensity and PGA and derive corresponding relationships between macroseismic intensity and PGA. The intensity-PGA relationship developed in this study is fully compatible with results of previous research.

## 1 Introduction

Field surveys after major disastrous earthquakes have shown that poor performance of buildings in earthquake affected areas is the leading cause of human fatalities and economic losses (Yuan, 2008). The evaluation of seismic fragility for existing building stocks has become a crucial issue due to the frequent occurrence of earthquakes in the last decades (Rota et al., 2010). Building fragility curves, defined as expected building damage under given earthquake ground shaking, have been developed for different typologies of buildings. They are required for the estimation of fatalities and monetary losses due to structural damage.

The development of fragility analyses can be divided mainly into two approaches: empirical methods and analytical methods. Empirical methods are based on post-earthquake surveys and considered to be the most reliable source, because they are directly correlated to the actual seismic behaviour of buildings (Maio and Tsionis, 2015). Numerous post-earthquake investigations have been conducted for groups of buildings to derive the empirical damage matrices. A damage matrix is a table of predefined damage states and percentages of a



specific building type at which each damage state is exceeded due to a particular macroseismic intensity level. However, empirical investigations are usually limited to particular sites or seismo-tectonic/geotechnical conditions with abundant seismic hazard and lack generality (Billah and Alam, 2014). Moreover, they usually refer to the macroseismic intensity, which is not an instrumental measure but is based on a subjective evaluation

(Maio and Tsionis, 2015). By contrast, analytical methods are based on static and dynamic nonlinear analyses of modelled buildings, which can produce slightly more detailed and relatively transparent assessment algorithms with direct physical meaning (Calvi et al., 2006), thus are conceived to be more reliable than empirical results (Hariri-Ardebili and Saouma, 2016). Nevertheless, variations in the following practices, such as selection of seismic demand inputs, employment of analysis techniques, characterisation of modelling structures, definition

of damage states thresholds as well as usage of damage indicators by different authorities, can create discrepancies among various analytical results even for exactly the same building typology.  In addition, analytical fragility studies for groups of buildings are computationally demanding and often technically difficult to perform.

In seismic risk assessment, the application of the existing fragility curves has been considered as a challenging task due to the fact that different approaches and methodologies are spread across scientific journals, conference proceedings, technical reports and software manuals, hindering the creation of an integrated framework that could allow the visualization, acquisition and comparison between all the existing curves (Maio and Tsionis, 2015). In this regard, the purpose of this study is to describe and examine available fragility curves, specially

developed for Chinese buildings from 87 papers using empirical and analytical methods. The median fragility results from these previous research findings for the main building types in China are then outlined. Empirical and analytical fragility curves are derived based on the median values. In addition, the two types of fragility curves allow the derivation of relationships between PGA and macroseismic intensity. Formally, this results from the elimination of the fragility values from the fragility–intensity and from the fragility–PGA relation.

Reasonable results should emerge if the building types used for analytic calculations and those used in the empirical field studies are close enough.

## 2    Fragility curves for Chinese buildings

In this paper we review 87 existing fragility analyses for the main building typologies in China, since building fragility constitutes one of the three main components of seismic risk assessment and loss mitigation. In China,

the main building types of concern are masonry and RC buildings, due to the wide distribution of masonry in rural and township areas and the increasing popularity of RC buildings in urban areas. As documented in Calvi et al. (2006), the first employment of empirical methodology to assess building fragility at large geographical scales was carried out in the early 1970s. In China, since the 1975 Haicheng M7.5 earthquake, around 112 post-earthquake surveys (Ding, 2016) have been conducted for Mag≥4.7 earthquakes occurred in seismic prone

provinces including **Sichuan** (Chen et al., 2017; Gao et al., 2010; He et al., 2002; Li et al., 2015; Li et al., 2013; Sun et al., 2013; Sun et al., 2014; Sun and Zhang, 2012; Ye et al., 2017; Yuan, 2008; Zhang et al., 2016), **Yunnan** (He et al., 2016; Ming et al., 2017; Piao, 2013; Shi et al., 2007; Wang et al., 2005; Yang et al., 2017; Zhou et al., 2007; Zhou et al., 2011), **Xinjiang** (Chang et al., 2012; Ge et al., 2014; Li et al., 2013; Meng et al., 2014; Song et al., 2001; Wen et al., 2017), **Qinghai** (Piao, 2013; Qiu and Gao, 2015), **Fujian** (Bie et al., 2010;



Zhang et al., 2011; Zhou and Wang, 2015) and **other seismic active zones** (Anaer, 2013; Chen, 2008; Chen et al., 1999; Cui and Zhai, 2010; Gan, 2009; Guo et al., 2011; Han et al., 2017; He and Kang, 1999; He and Fu, 2009; He et al., 2017; Hu et al., 2007; Li, 2014; Liu, 1986; Lv et al., 2017; Ma and Chang, 1999; Meng et al., 2012; Meng et al., 2013; Shi et al., 2013; Sun and Chen, 2009; Sun, 2016; Wang et al., 2011; Wang, 2008; Wang, 2007; Wei et al., 2008; Wu, 2015; Xia, 2009; Yang, 2014; Yin et al., 1990; Yin, 1996; Zhang and Sun, 2010; Zhang et al., 2017; Zhang et al., 2014; Zhou et al., 2013). One main output of these post-earthquake surveys are empirical damage probability matrices (DPMs), which depict the discrete conditional probability of exceeding predefined damage limit states referred to different macroseismic intensity levels. That is, for the DPMs, macroseismic intensity is usually used as the ground motion indicator. The detailed definition of each intensity level is written in Section 4.1 of the Chinese Official Seismic Intensity Scale (GB17742-2008) (Appendix Table B1).

As aforementioned, the main drawback of empirical method lies in the subjectivity on allocating each building to a damage state or in the lack of accuracy in the determination of the ground motion affecting the region (Maio and Tsionis, 2015). Furthermore, the interdependency between macroseismic intensity and damage and the limited or heterogeneous empirical data are commonly identified as the main difficulties to overcome in the calibration process of empirical approaches (Del Gaudio et al., 2015).

By contrast, analytical methodologies produce slightly more detailed and transparent algorithms with direct physical meaning, that not only allow detailed sensitivity studies to be undertaken, but also cater for the straightforward calibration to various characteristics of the building stock and seismic hazard (Calvi et al., 2006). Different from the empirical fragility that is directly collected from post-earthquake survey, the derivation of analytical fragility curves is often based on nonlinear fine-element analysis. Popular analytical methods include push-over analysis (Freeman, 1975; Freeman, 1998), adaptive push-over method (Antoniou et al., 2002; Antoniou and Pinho, 2004), and incremental dynamic analysis (IDA) (Vamvatsikos and Cornell, 2002; Vamvatsikos and Fragiadakis, 2010). Within these approaches, most of the methodologies available in literature rest on two main and distinct procedures: the correlation between acceleration or displacement capacity curves and spectral response curves, as the well-known HAZUS or N2 methods (FEMA 2003; Fajfar 2000), and the correlation between capacity curves and acceleration time histories, as proposed in Rossetto and Elnashai (2005). The major steps in performing analytical methods include: the selection of seismic demand inputs, the construction of building models, the selection of damage indicator and the determination of damage limit state criteria, as illustrated in Fig. 1 (Dumova-Jovanoska, 2000). To combine empirical post-earthquake damage statistics with simulated, analytical damage statistics from modelled building typology under consideration, we examined studies deriving analytical fragility curves for masonry and RC buildings in China. The analysis techniques in these studies vary from static push-over analysis or adaptive push-over method (Cui and Zhai, 2010; Liu, 2017), to dynamic history analysis or incremental dynamic analysis (Zeng, 2012; Liu et al., 2010; Liu, 2014; Liu, 2014; Sun, 2016; Wang, 2013; Yang, 2015; Yu et al., 2017; Zheng et al., 2015; Zhu, 2010) as well as based on some statistical assumptions (Fang, 2011; Gan, 2009; Guo et al., 2011; Hu et al., 2010; Zhang and Sun, 2010).

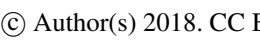



As defined beforehand, building fragility describes the exceedance probability of specific damage state given an ensemble of earthquake ground motion levels. To describe the susceptibility of building structure to certain ground motion level, four damage limit states are used to discriminate between different strengths of ground shaking: slight damage (LS1), moderate damage (LS2), serious damage (LS3) and collapse (LS4). These four limit states divide the structure into five structural damage states, namely negligible (D1), slight damage (D2), moderate damage (D3), serious damage (D4) and collapse (D5). The relationship between limit states and structural damage states is illustrated by Fig. 2.

Detailed descriptions of building structural damage states have been issued in different countries and areas. In the European Macroseismic Scale 1998 (EMS-98) proposed by European Seismological Commission (ESC), five grades of structural damage are defined: negligible to slight damage (Grade 1), moderate damage (Grade 2), substantial to heavy damage (Grade 3), very heavy damage (Grade 4) and destruction (Grade 5). In the HAZUS99 Earthquake Model Technical Manual, developed by Department of Homeland Security, Federal Emergency Management Agency of the United States (FEMA) in 1999, generally four structural damage classes are used for all building types: slight damage, moderate damage, extensive damage and complete damage. Other damage state classifications like MSK1969 proposed by Medvev and Sponheuer (1969) and AIJ1995 in Japan issued by Architectural Institute of Japan are summarized in Table 1. In this work, we referred to the damage states elaborated in China's official standard proposed by China Earthquake Administration (CEA). The latest standard, in which damage details for structural and non-structural components are defined for each damage state was issued in 2008 (GB17742-2008: The Chinese seismic intensity scale), as can be seen from Table 2.

In post-earthquake field investigation, macroseismic intensity is usually used as the indicator of ground motion. The macroseismic intensity level is derived from the damage state of specific building type. In China, three types of building are used to determine earthquake intensity: (1) **Type A**: wood-structure, soil/stone/brick-made old building; (2) **Type B**: single- or multi-storey brick masonry without seismic resistance; (3) **Type C**: single- or multi- storey brick masonry with VII seismic resistance. The detailed building structural damage description for judgement of each intensity scale can be referred to Table B2 (a non-official translation based on the currently latest version of China seismic intensity scale: GB17742-2008).

Fragility curves depict the exceedance probability of each damage limit state (LS1, LS2, LS3, LS4) given a specified level of ground motion. In the empirical fragility analysis method, fragility curve can be directly derived from damage probability matrices (DPMs) that determined from post-earthquake field surveys for each structural damage state. DPMs give the proportions of buildings in each damage state (D1, D2, D3, D4, D5), thus can be used to derive the probability of exceeding each damage limit state $P[LS_i]$ (i=1,2,3,4), as illustrated in Eq. (1):

$$P[LS_i] = 1 - P[D_i] \ (i = 1); \quad P[LS_i] = P[LS_{i-1}] - P[D_i] \ (i = 2 \dots N) , \qquad (1)$$

where $N$ refers to the total number of damage limit states (here $N$=4); $P[D_i]$ refers to the proportion of specific building type in each damage state $i$.

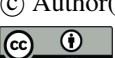



In analytical fragility method, building response to seismic demand inputs is assumed to follow the lognsormal distribution in Eq. (2):

$$P[LS|S_d] = \Phi\left[\frac{1}{\beta_{LS}}\ln(\frac{S_d}{S_{C|LS}})\right],$$ (2)

where $P[LS|S_d]$ is the probability of being in or exceeding damage limit state *LS* due to ground motion indicator

$S_d$; $S_{C|LS}$ refers to the median value of damage state indicator at which the building reaches the threshold of the damage state *LS*; $\beta_{LS}$ represents the overall uncertainties resulting from seismic demand input, building capacity and model uncertainty. The overall uncertainty is generally within the range of 0.6-0.8 (Der Kiureghian and Ditlevsen, 2009; Wenliuhan, 2015); $\Phi[\ ]$ is the standard normal cumulative probability distribution function.

### 3    Analysis of existing fragility curves

During the past four decades, around 112 Mag≥4.7 damaging earthquakes have information for the densely populated areas in mainland China since the Haicheng M7.5 earthquake in 1975 up to the Ludian M6.5 earthquake in 2014 (Ding, 2013). These earthquakes mainly occurred in seismic prone provinces in western China (e.g. Xinjiang Uygur, Tibet, Qinghai) and southwestern China (e.g. Sichuan, Yunnan). The main building types in these areas are featured by masonry, reinforced concrete (RC), brick-wood, soil, stone as well as

chuandou-timber (a typical building type in mountainous area of Tibetan, Qinghai and Sichuan). Due to the data abundance, we mainly focus on the seismic vulnerability of the two most widely distributed building types in China: masonry and RC buildings. Masonry buildings are mainly composed of brick and concrete. RC buildings include building structures such as RC core wall, frame structure, frame-shear wall. The seismic resistance level of masonry and RC buildings is further divided into two classes: level A and level B. The assignment of seismic

resistance level of buildings in different studies is mainly based on the building construction age and corresponding code level given in the literature (Table 3), supplemented by the location and structure material information of damaged buildings. Generally "level A" includes buildings with seismic resistance level assigned as pre/low/moderate-code, and "level B" indicate buildings assigned as high-code seismic resistance level.

After grouping the empirical damage probability matrices and analytical fragility data based on building type (masonry and RC) and seismic resistance level (A and B) (*these data are accessible through* https://www.jianguoyun.com/p/DdSVac8QgPb4BhiHhnY), the overall empirical fragility based on macroseismic intensity (Fig. 3) and analytical fragility based on PGA (Fig. 4) for four damage limit states (LS1, LS2, LS3, LS4) are derived. In this work, "fragility" refers to the exceedance probability of each damage limit

state at each ground motion level. In Fig. 3 and Fig. 4, the scatter of fragility varies over building types and seismic resistance levels. For empirical fragilities, the scatter may be due to the limited abundance of damage data for buildings investigated in post-earthquake field surveys, the subjective judgement of damage states among different authorities around China as well as the rough division of building structure types and seismic resistance levels. For analytical fragilities, the scatter may come from the difference in the selection of seismic

demand inputs, the usage of analysis techniques, the detailing of the modelled building structure, the definition of damage state as well as the difference in damage indicators used by different researchers. Thus before deriving building fragility curves using these discrete datasets, a box-plot check method was performed to remove the outliers in these original fragility data.





Considering the scatter in original empirical and analytical datasets, the box-plot check method was firstly used to remove the outliers. For each building type (masonry_A, masonry_B, RC_A, RC_B) and in each damage limit state (LS1, LS2, LS3, LS4), the corresponding series of fragility data was sorted from the lowest to the highest value. Three quantiles ($Q_1$, $Q_2$, $Q_3$) were used to divide each fragility series into four equal-sized groups and they correspond to the 25%, 50% and 75% quantile value in each series. A discrete fragility value ($Q_i$) was assigned as an outlier if $Q_i - Q_3 > 1.5 \times (Q_3 - Q_2)$ or $Q_1 - Q_i > 1.5 \times (Q_2 - Q_1)$. The box-plot check results for empirical fragility data (Fig. 5) and analytical fragility data (Fig. 6) are as follows.

## 4    Derivation of representative fragility curves

After removing outliers, by using the median fragility values, namely the 50% quantile fragility value of each fragility series, median fragility curves can be derived for four damage limit states (LS1, LS2, LS3, LS4) of four building types (masonry_A, masonry_B, RC_A, RC_B). For this purpose we match the 2 parameters μ and σ in the cumulative normal or log-normal distribution for the empirical fragility datasets (Fig. 7) and analytical fragility datasets (Fig. 8). For each damage limit state of each building type, the regression parameters $\mu_{LS}$ and $\sigma_{LS}$ can be fitted using the expression in Eq. (3):

$$P(X|LS) = \Phi\left[\frac{1}{\sigma_{LS}}\ln\left(\frac{X_{PGA}}{\mu_{LS}}\right)\right] \quad or \quad P(X|LS) = \Phi\left[\frac{X_{int}-\mu_{LS}}{\sigma_{LS}}\right], \tag{3}$$

where $P(X|LS)$ represents the exceedance probability of each damage limit state $LS$ given ground motion level X (X refers to $X_{PGA}$ for PGA in analytical fragility and $X_{int}$ for intensity in empirical fragility).

As can be clearly seen in both Fig. 7 (empirical fragility curves) and Fig. 8 (analytical fragility curves), there are two obvious trends: (1) for the same building type (masonry or RC), the higher the seismic resistance level (A<B), the lower the building fragility, which applies for all damage limit states; (2) for the same seismic resistance level, RC building has lower fragility than masonry building, which also applies for all damage limit states. These indicate the reliability of our original fragility data collected and the reasonability of our criteria in grouping building types and seismic resistance levels.

Mathematically, the fitness accuracy between the derived fragility curve and the original discrete fragility dataset can be measured from statistical indicators such as the R-squared value (Draper and Smith, 1998). Higher R-squared values indicate a better fit for the fragility curve regression. Since the R-squared value is defined as the ratio between *SSR* and *SST, SSR* is the *sum of squares of the regression* ($SSR = \sum_{i=1}^{n}(\hat{y}_i - \bar{y}_i)^2$), and SST is the *total sum of squares* ($SST = \sum_{i=1}^{n}(y_i - \bar{y}_i)^2$); $y_i$ refers to the original discrete fragilities for each limit state, $\bar{y}_i$ refers to the mean fragility, $\hat{y}_i$ refers to the predicted fragility by the fitted fragility curve. Detailed information (e.g., the number of data points) of fragility values used in deriving fragility curve for each damage state of each building type is given in Table B1.

As shown in Table 4, the R-squared values are generally above 0.95, which indicates the normal or lognormal distribution assumption in Eq. (3) is very suitable to match the discrete fragility datasets. There are also three noticeably low R-square values (≤0.8) in Table 4 for damage limit states LS1, LS2, LS3 of building type "RC_A". Fig. 4 and even better Fig. 6 show that the originally collected analytical fragility data for "RC_A" are





more scattered than for other building types. This thus leads to the low R-square values for damage limit states LS1, LS2, LS3 of "RC_A".

## 5  Intensity-PGA relationships

Traditionally, intensity-PGA relationships are developed using PGA records from instrumental observations and macroseismic intensity observations from damage surveys within the same geographical range. These relationships are generally region-dependent and have large scatter (Caprio et al., 2015). In this study, we derive fragility curves using the empirical and analytical datasets described in this paper. For each building type and each damage limit state, an empirical fragility curve (exceedance probability vs. macroseismic intensity) and an analytic fragility curve (exceedance probability vs. PGA) are available. By eliminating the fragility values we can derive relationships between macroseismic intensity and PGA. In previous practices in fitting intensity and PGA relationship ( Bilal and Askan, 2014; Caprio et al., 2015; Ding et al., 2014; Ding, 2016; Ding et al., 2017; Ogweno and Cramer, 2017; Worden et al., 2012), PGA and intensity datasets chosen to fit are mainly because these records are within the same geographical range. But no further classification of datasets was considered based on building type and damage state as here in this study. This lack of detailed PGA and intensity datasets classification before regression may well explain why the previously derived intensity-PGA relationships generally had high scatter and obvious regional dependence (Caprio et al., 2015). Here we derive regression relationships between intensity and PGA for each damage limit state of each building type separately. Theoretically all the intensity-PGA relationships should coincide regardless of the building type and damage limit state. Using Eq. (3) for PGA-fragility and intensity-fragility respectively and eliminating fragility as variable, we find:

$$\ln(PGA) = \alpha + \beta * Int,$$

$$with\ \alpha = \ln(\mu_{PGA}) - \frac{\sigma_{PGA}}{\sigma_{Int}} * \mu_{Int},\ \beta = \frac{\sigma_{PGA}}{\sigma_{Int}}, \tag{4}$$

In which, the parameters $\mu_{PGA}$, $\mu_{Int}$, $\sigma_{PGA}$, $\sigma_{Int}$ are taken from Table 3 with different values dependent on building type and damage limit state.

The relationships between macroseismic intensity and PGA derived are plotted in Fig. 9 (grouped by building types) and Fig. 10 (grouped by damage limit states). Higher damage states can occur only for higher intensities or PGA values. For instance, a LS4 damage state at intensity III would not happen. Thus, we plot the intensity-PGA relations only for fragility values above 1%. As a consequence, the curves in Fig. 9 and Fig. 10 for LS1 have the lowest PGA or intensity starting point, while LS4 has the highest. Ideally, we would expect the overlap of all relationship curves between macroseismic intensity and PGA, regardless of the grouping method, whether by building type or by damage state. In reality, the four intensity-PGA curves coincide very well for the four damage limit states within building type "masonry_A" and "masonry_B" in Fig. 9. Meanwhile, the noticeable discrepancy in intensity-PGA relationships of "RC_A" for damage states LS1, LS2, LS3 in Fig. 9 is not surprising, given the relatively high scatter in the original analytical fragility datasets of "RC_A" (as can be further checked from Appendix Fig. A1-A4).





In addition, for building type "RC_A" and "RC_B" in Fig. 9, given the same macroseismic intensity, the corresponding PGA values in damage state LS4 are much higher than in the other three damage states LS1, LS2, LS3. If we look back to the original data collection work, since the derivation of empirical fragility curve requires the collection of post-earthquake damage statistics at sites with similar ground conditions for a wide

range of ground motion levels, which often mean that the statistics from multiple earthquake events need to be combined. Furthermore, large magnitude earthquakes occur relatively infrequently near densely populated areas and so the data available tends to be clustered around the low damage state and ground motion levels, thus limiting the validation of high damage state or ground motion levels (Calvi, 2006). Besides, the seriously damaged buildings in earthquake affected area in China are mainly masonry. Thus for RC_A and RC_B, the

scarcity of damage data for high damage state or ground motion level may explain why intensity-PGA regression relationships for damage states LS4 of "RC_A" and "RC_B" are abnormal. Therefore we believe in Fig. 9, the abrupt high PGA of LS4 especially in building type "RC_B" can be well explained by the scarce data availability (as can be checked out from Appendix Table B1).

Due to the original damage data scarcity and high scatter for certain building types and damage limit states as aforementioned, we discard the less reliable intensity-PGA relationships of LS1, LS2, LS3, LS4 in "RC_A" and LS4 in "RC_B" as in Fig. 9. The remaining regression curves within "masonry_A", "masonry_B" and "RC_B" coincide quite well. Only that in "masonry_B", the PGAs revealed by its four damages state are generally higher than that in "masonry_A" given the same intensity level. This can be more clearly seen in Fig. 10, in which the

intensity-PGA relationships are grouped by damage limit states. That is, for each limit state, the relationships revealed by all four building types are plotted together. Given the same intensity level, the PGAs revealed by "masonry_B" are generally higher than that by all the other three building types. How to explain this abnormal phenomenon then?

Actually, compared with "masonry_A", the "masonry_B" building type generally has high seismic resistance level or anti-earthquake capacity. Thus for the same macroseismic intensity level, the damage posed on "masonry_B" should be slighter than on "masonry_A". In post-earthquake filed surveys, the standard way in deriving macroseismic intensity level in China is by visually checking the damage states of three officially assigned buildings types, namely (1) **Type A**: wood-structure, soil/stone/brick-made old building; (2) **Type B**:

single- or multi-storey brick masonry without seismic resistance; (3) **Type C**: single- or multi- storey brick masonry with VII seismic resistance. In this study, the seismic resistance level we assigned to "masonry_B" buildings is generally higher than all the three officially referred Type A/B/C buildings. Thus intensity levels derived from the damage of "masonry_B" buildings in post-earthquake survey should be relatively lower than from officially referred Type A/B/C buildings. This may explain why given the same PGA level, the

corresponding intensity revealed by "mansory_B" is lower than that in "mansory_A". Or vice versa, given the same macroseismic intensity level, the corresponding PGA revealed by "masonry_B" is thus higher than that by "masonry_A".

For building type "masonry_A", the intensity-PGA curves are quite similar for all damage states, although for

reasons explained earlier that the higher damage states cover only higher values of intensities and PGA values. If

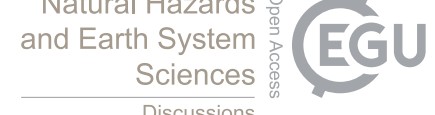



we average the curves for discrete intensity values, we derive the corresponding averaged PGA values as listed in Table 5.

If we match the data points in Table 5 with a linear relationship between intensity and ln(PGA), we find Eq. (5):

$$\ln(PGA) = 0.521 * Int - 5.43 \pm \varepsilon \qquad (PGA: \text{g}) \tag{5}$$

where ε follows the normal distribution, with 0 as the median value and the standard deviation is σ.

Considering the scatter of "masonry_A" in both empirical fragility data and analytical fragility data, the averaged standard deviation σ in Eq. (5) is estimated as 0.3. As the "masonry_A" type is the most abundant building type and the most relevant one for historic earthquakes, we recommend using Eq. (5) for the building damage assessment of historical earthquakes. Moreover, if we only remove those several uncertain intensity-PGA curves as analyzed beforehand, namely LS1, LS2, LS3, LS4 of "RC_A" and LS4 of "RC_B", the PGA values derived from the remaining intensity-PGA relationships have broader ranges, as listed in Table 6.

In the Chinese Official Seismic Intensity Scale (GB17742-2008), the recommended PGA range for each intensity level is listed in Table 7. The PGA values for intensity VI, VII in our results are higher than that given in Table 7; for intensity VII, IX and X, the PGA values compare quite well with each other.

The recommended PGA ranges for each intensity level in GB17742-2008 are the same as those given around three decades ago in GB17742-1980. The Chinese damage data used to derive the intensity-PGA relation in 1980 was scarce. Thus damaging earthquakes occurred before 1971 in the United States were also largely used, which may not be representative of the situation in China today. Therefore we also compared our results with the latest intensity-PGA relation (Ding, 2017) developed based on the strong ground motion records and damage reports for 28 Mag≥5 earthquakes occurred in mainland China during 1994-2014. The corresponding PGA values for intensity VI-IX given in Ding (2017) are listed in Table 8.

Comparing our results in Table 5, 6 with that in Table 8, it can be seen that the PGA values, though developed by different methods, are quite consistent for both low intensity (VI, VII) and high intensity (VIII, IX) levels. A possible explanation for the relatively low PGAs for low intensity level (VI, VII) in Table 7 is that, the building fragilities in the 1980s were higher than todays' building stock. Since macroseismic intensity is a direct indicator of building damage, todays' building stocks generally require higher ground motion (PGA) than buildings in 1980s to cause that same damage.

## 6 Conclusion

We evaluate 69 papers, mostly from the Chinese literature, that document observations of macroseismic intensities reflecting earthquake damage in all seismic events of M≥4.7 earthquakes occurred in densely populated areas in mainland China over the past four decades. The papers provide empirical fragilities for 4 damage limit states (LS1, LS2, LS3, LS4) dependent on intensities for 4 building/construction types (masonry_A, masonry_B, RC_A, RC_B). From this wealth of data, we derive the median fragility curves for these building





types by removing outliers and deem them to be the most representative fragilities of the various functions. We also discuss the uncertainties and these representative fragilities should be very valuable for loss modeling when in terms of macroseismic intensities.

We also scrutinize 18 papers with results on modeling fragilities for the same building/construction types and the same damage classes either by response spectral methods or by time-history response analysis. These analytic methods provide fragilities as functions of PGA. Again, we remove outliers and derive the median fragilities of the many studies including the uncertainties. Also these representative fragilities are valuable for loss modeling based on the engineering ground motion parameter PGA.

We compare both streams of information by deriving intensity-PGA relationships independently for each building/construction type and each damage class. Ideally the individual intensity-PGA relations should all coincide and allow us to derive an average relation between intensity and PGA. The coincidence is not 100% perfect and deviations for the cases where they occur are discussed. For studies referring to historic earthquakes and their losses we recommend utilizing the relationship for "masonry_A" buildings in Eq. (5).

We mention some limitations of the results of this paper. First, the classification of building types and seismic resistance levels are kind of simplified to get a solid database for each specific building type. Second, the discrete damage datasets we used to derive fragility curve are essentially the median fragility for each intensity or PGA level, instead of using the averaged damage matrix. Third, the range of buildings used for intensity determination and for analytical studies do not coincide. A "masonry_A" building in a post-event field survey may encompass a wider range than in an analytic study. As – in the end – the representative median fragilities are considered, this may not be a relevant short-coming.

**Appendix**

To explore the scatter of the original fragility datasets for each limit state of each building type, we performed the error-bar analysis, as shown in Fig. A1 (empirical data) and Fig. A2 (analytical data). Specifically, to better scale the scatter, standard deviations of fragility, namely the exceedance probability of each damage limit sta**te** are plotted in Fig. A3 (empirical data) and Fig. A4 (analytical data), respectively. Detailed dataset information including the number of fragility data before and after removing the outliers, median fragility value used in deriving fragility curve in Fig. 7 and Fig. 8 and the standard deviation in each dataset can be referred to Table B1.

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



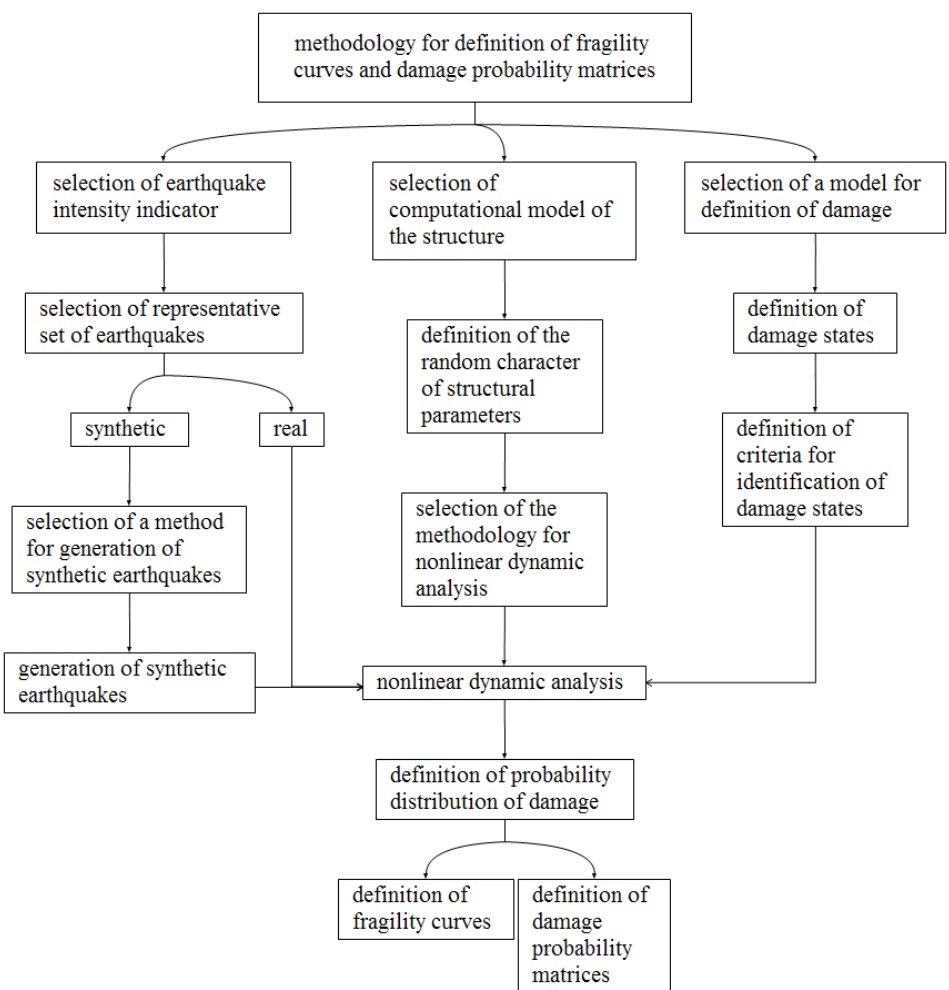

**Figure 1: Flowchart to describe the components of the calculation of analytical fragility curves and damage probability matrices (modified after Dumova-Jovanoska, 2000).**

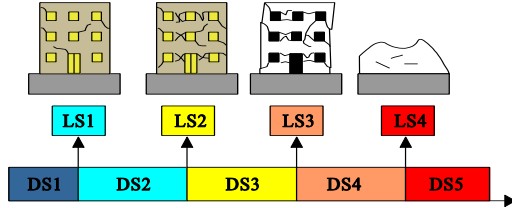

**Figure 2: Corresponding Relation between structural damage states (DS1, D2, D3, DS4, DS5) and limit states (LS1, LS2, LS3, LS4) (modified after Wenliuhan, 2015)**



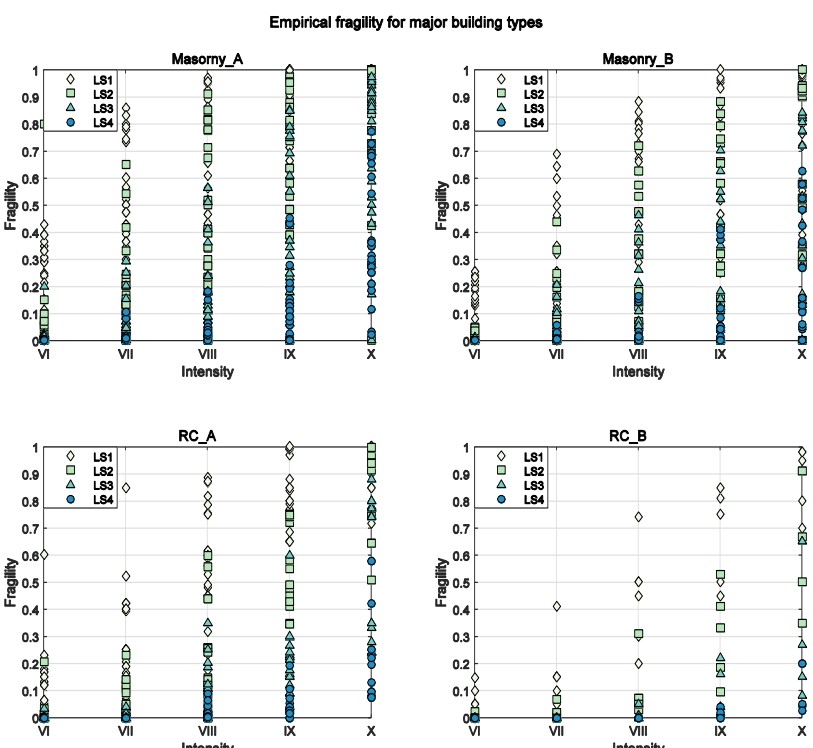

**Figure 3: The distribution of empirical fragility data from post-earthquake field surveys, depicting the relationship between the exceedance probability of each damage limit state (LS1, LS2, LS3, LS4) at given macroseismic intensity levels. The fragility datasets are grouped by building types (masonry and RC) and seismic resistance levels (A and B).**





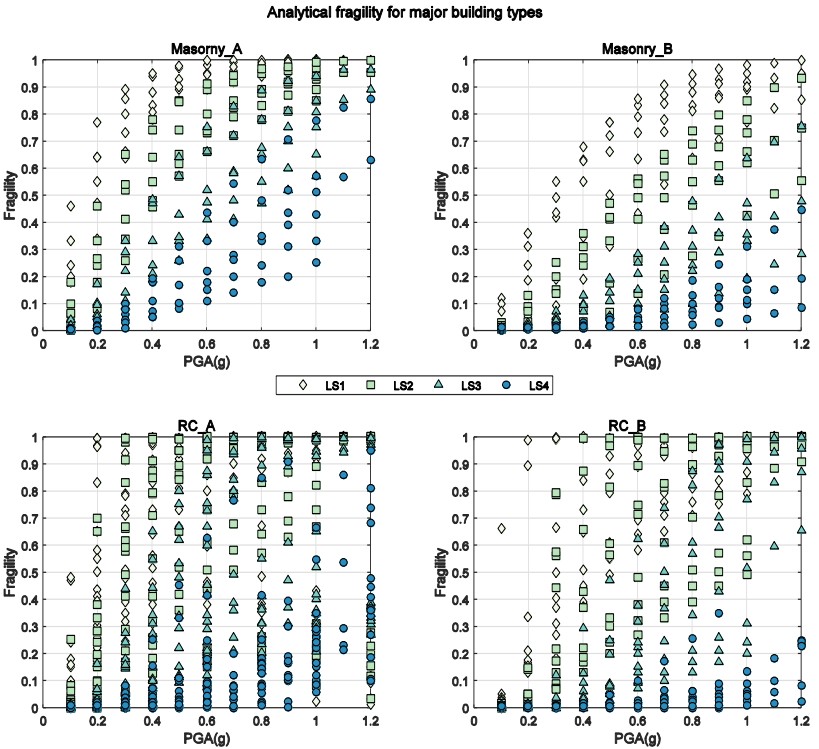

**Figure 4: The distribution of analytical fragility data derived from non-linear analyses, depicting the relationship between the exceedance probability of each damage limit state (LS1, LS2, LS3, LS4) at given PGA levels. The fragility datasets are grouped by building types (masonry and RC) and seismic resistance levels (A and B).**




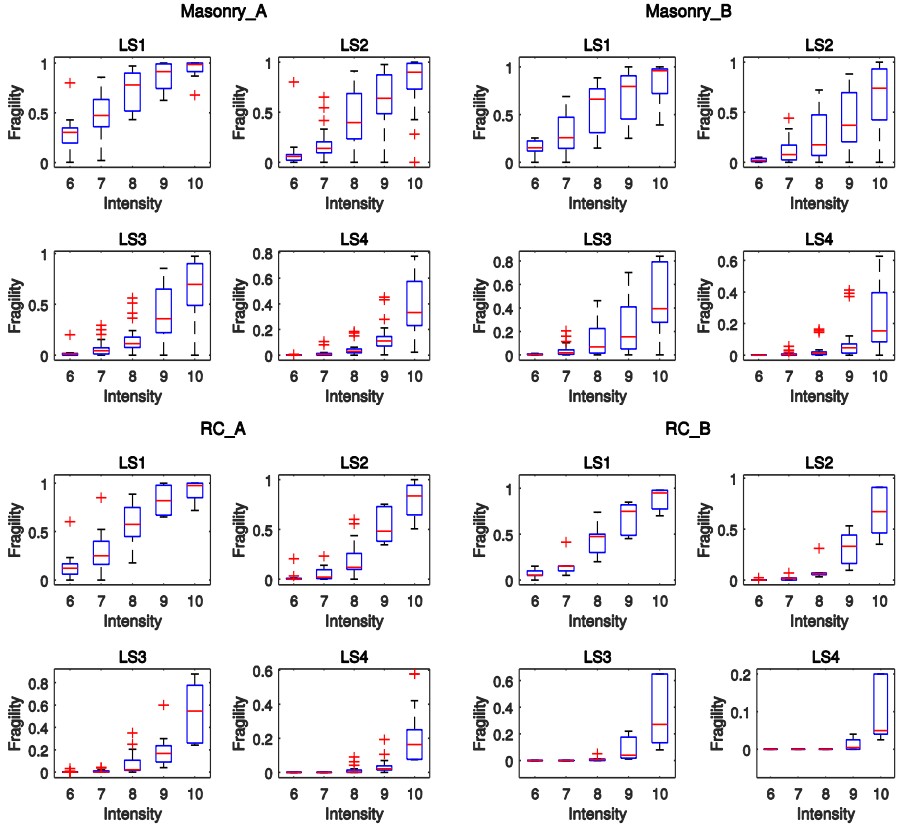

**Figure 5: Outlier-check using box-plot method for empirical fragility data. Five macroseismic intensity levels are used to group the original fragility datasets: VI, VII, VIII, IX, X. "A" and "B" represent the pre/low/moderate-code and high-code seismic resistance level, respectively. LS1, LS2, LS3, LS4 are the four damage limit states. Outliers are marked by red crosses.**





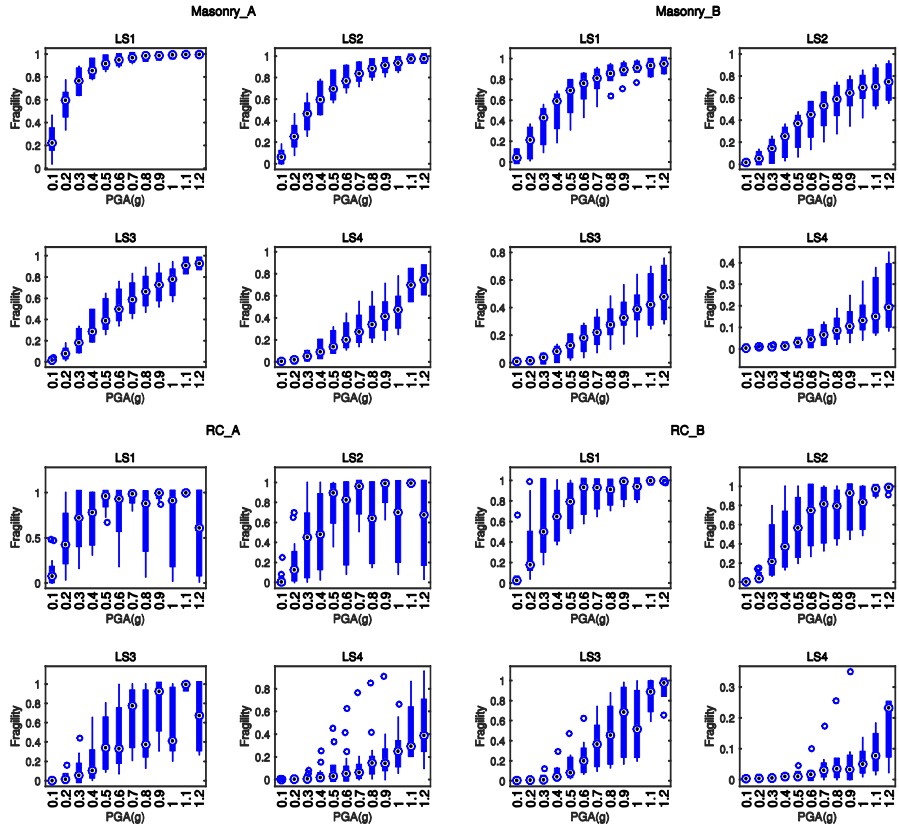

**Figure 6: Outlier-check using box-plot method for analytical fragility data. Twelve PGA levels are used to group the discrete analytical fragility datasets: 0.1-1.2 g. "A" and "B" represent the pre/low/moderate-code and high-code seismic resistance level, respectively. LS1, LS2, LS3, LS4 are the four damage limit states. Outliers are marked by blue hollow circles.**





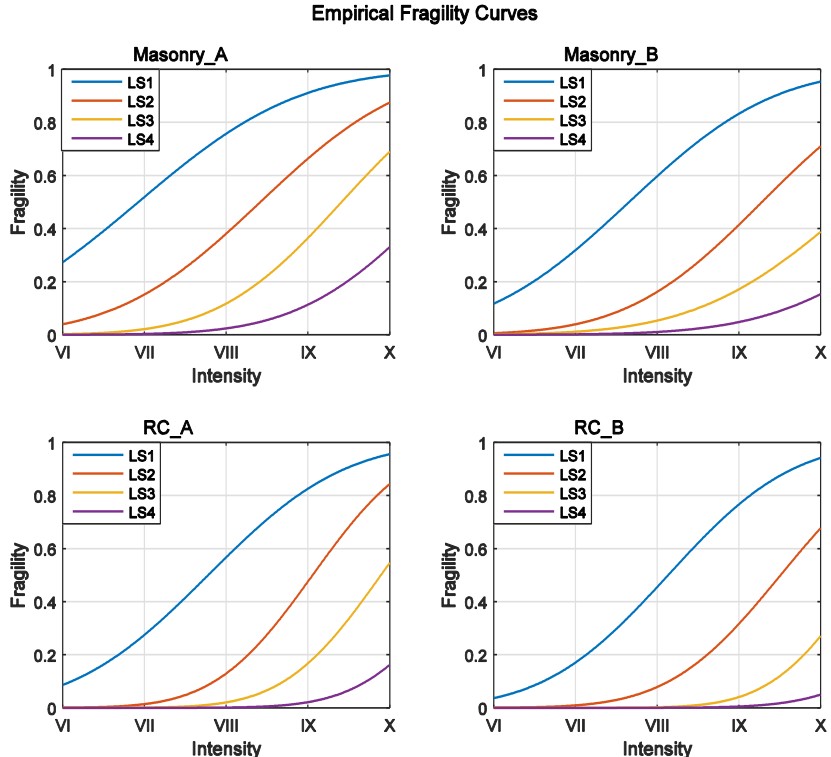

**Figure 7: Fragility curves derived from empirical fragility datasets, which depict the relationship between macroseismic intensity and exceedance probability of each damage limit state (LS1, LS2, LS3, LS4) for masonry and RC building types. The seismic resistance level "A" and "B" represent pre/low/moderate-code and high-code seismic resistance levels, respectively.**




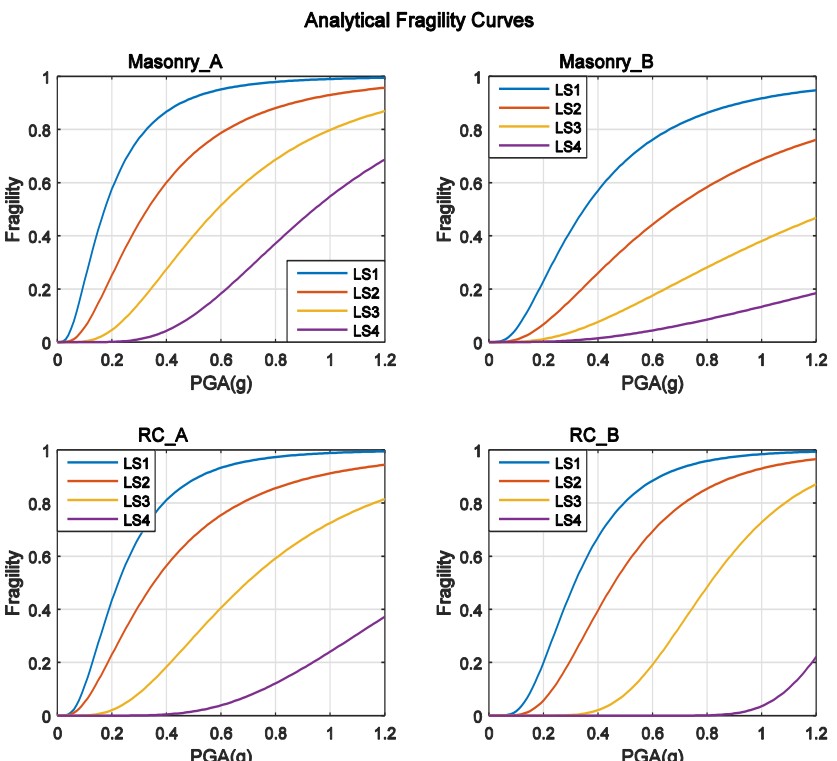

**Figure 8: Fragility curves derived from analytical fragility datasets, which depict the relationship between PGAs (unit: g) and exceedance probability of each damage limit state (LS1, LS2, LS3, LS4) for masonry and RC building types. The seismic resistance level "A" and "B" represent pre/low/moderate-code and high-code seismic resistance levels, respectively.**



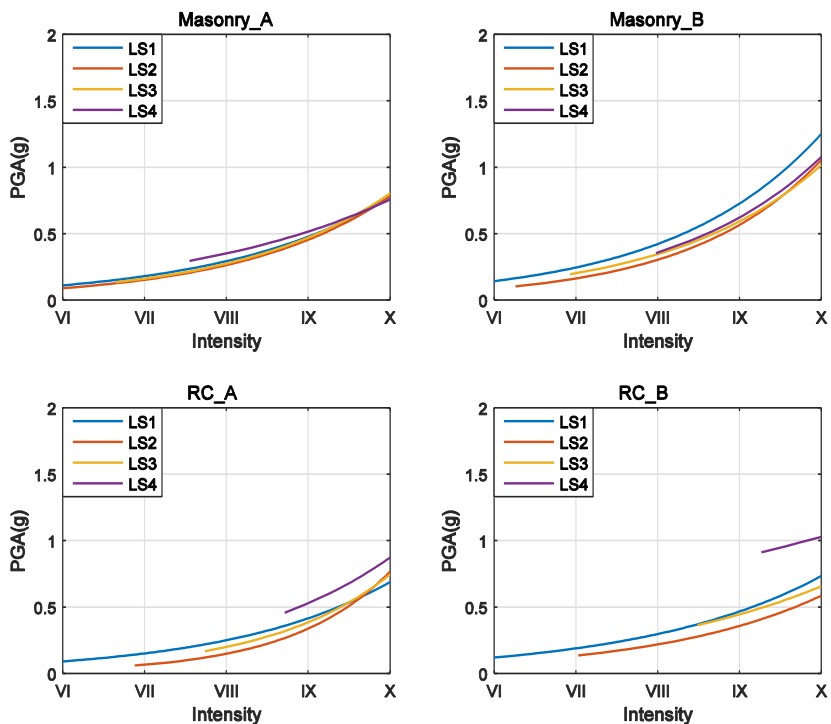

**Figure 9: Intensity-PGA relationships grouped by building types. Only intensity and PGA values with truncated exceedance probability ≥1% for each damage limit state of each building type are plotted, since higher damage states can appear only for higher intensities or PGA values.**



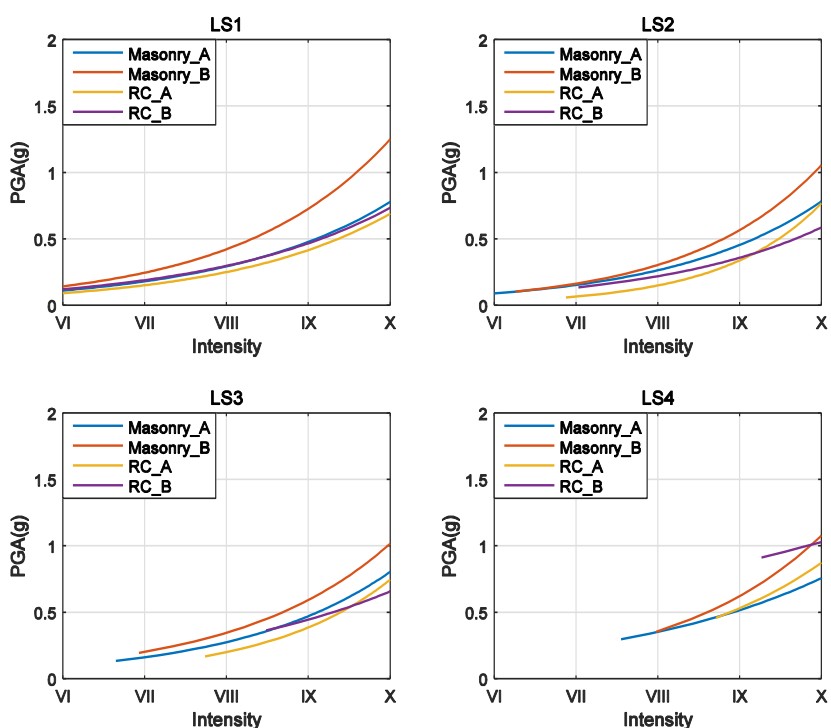

**Figure 10: Intensity-PGA relationships grouped by damage limit states. Only intensity and PGA values with truncated exceedance probability ≥1% for each damage limit state of each building type are plotted, since higher damage states can appear only for higher intensities or PGA values.**



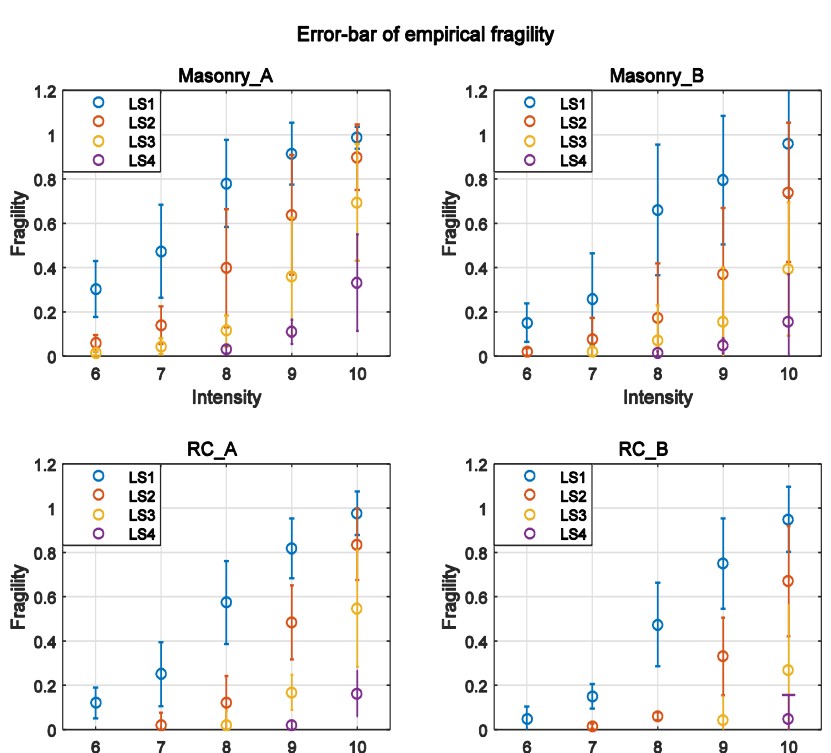

**Figure A1: The error-bar of empirical fragility, namely the exceedance probability of each damage limit state (LS1, LS2, LS3, LS4) for each building type (masonry_A, masonry_B, RC_A, RC_B) that derived from empirical fragility datasets. The circle within each bar represents the median exceedance probability of each damage limit state; the length of each bar indicates the value of the corresponding standard deviation. Only intensity and PGA values with truncated exceedance probability ⩾1% for each damage limit state of each building type are plotted, since higher damage states can appear only for higher intensities or PGA values.**





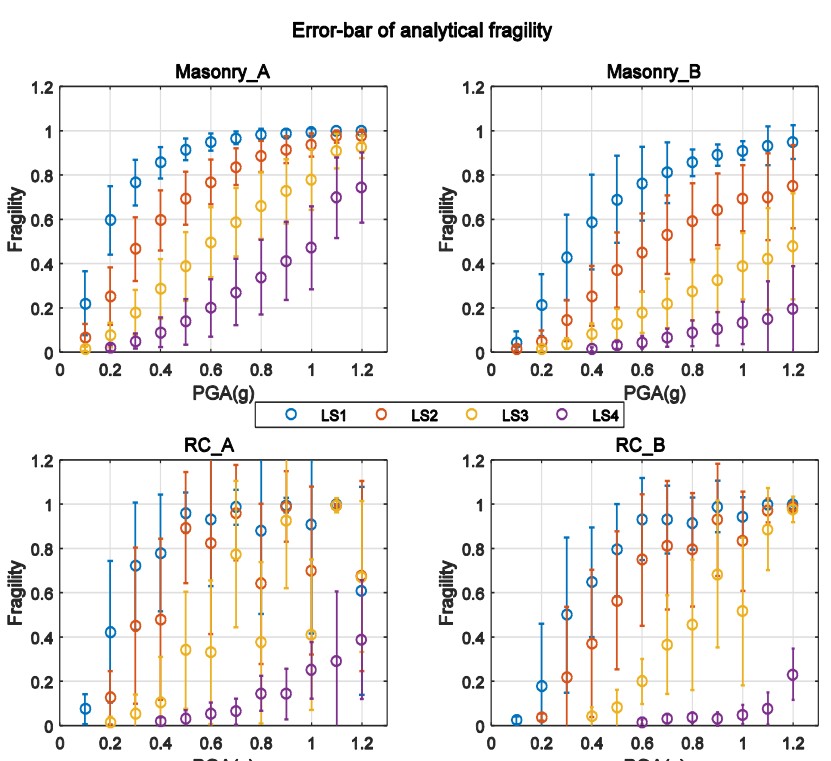

**Figure A2: The error-bar of analytical fragility, namely the exceedance probability of each damage limit state (LS1, LS2, LS3, LS4) for each building type (masonry_A, masonry_B, RC_A, RC_B) that derived from analytical fragility datasets. The circle within each bar represents the median exceedance probability of each damage limit state; the length of each bar indicates the value of the corresponding standard deviation. Only intensity and PGA values with truncated exceedance probability ≥1% for each damage limit state of each building type are plotted, since higher damage states can appear only for higher intensities or PGA values.**





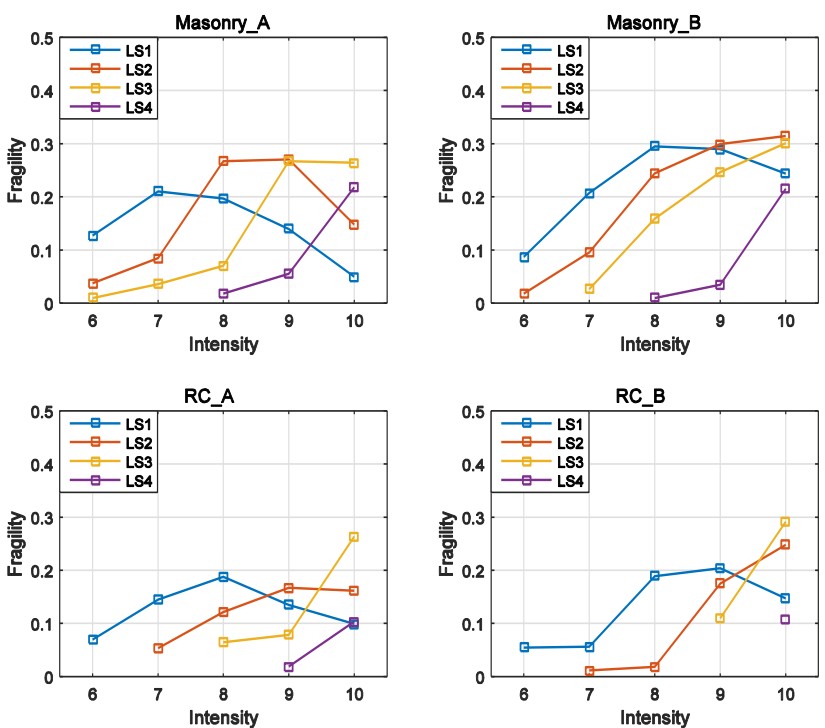

**Figure A3: Standard deviation of empirical fragility, namely the exceedance probability of each damage limit state(LS1, LS2, LS3, LS4) for each building type (masonry_A, masonry_B, RC_A, RC_B) that derived based on empirical fragility datasets. Only intensity and PGA values with truncated exceedance probability ≥1% for each damage limit state of each building type are plotted, since higher damage states can appear only for higher intensities or PGA values.**





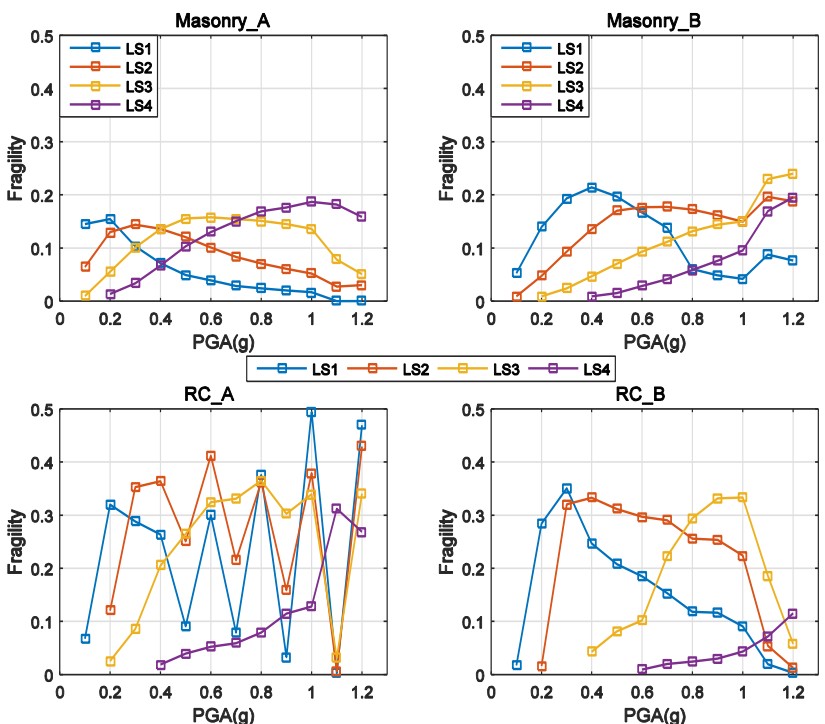

**Figure A4: Standard deviation of analytical fragility, namely the exceedance probability of each damage limit state(LS1, LS2, LS3, LS4) for each building type (masonry_A, masonry_B, RC_A, RC_B that derived based on analytical fragility datasets. Only intensity and PGA values with truncated exceedance probability ≥1% for each damage limit state of each building type are plotted, since higher damage states can appear only for higher intensities or PGA values.**



**Table 1. Example of major damage states classification methods (modified after Rossetto and Elnashai, 2003).**

| vulnerability | HAZUS1999 | EMS1998 | MSK1969 | AIJ1995 | China2008 |
|---|---|---|---|---|---|
| 0 | no damage | | | | |
| 10 | slight damage | Grade 1 | D1 | Light | D1 |
| 20 | | | | | D2 |
| 30 | | Grade 2 | D2 | Minor | D3 |
| 40 | | | | | |
| 50 | moderate damage | Grade 3 | D3 | | |
| 60 | | | | Moderate | D4 |
| 70 | | | | | |
| 80 | extensive damage | Grade 4 | D4 | Major | |
| 90 | | | | | D5 |
| 100 | complete damage | Grade 5 | | Partial collapse | |

**Table 2. Detailed definition of building damage states in GB17742-2008, China**

| Damage state | Structural damage | Non-structural damage | Performance-based Description |
|---|---|---|---|
| D1 | Negligible | Cracks only in *very few* non-structural components | No need to repair, instant use |
| D2 | *Very few* components have visible cracks | Obvious cracks can be found | No need to repair or after slightly repairing, can be used directly |
| D3 | *A few* components have slight cracks, *very few* have obvious cracks | *Most* components have serious damage | Certain repair work should be done before continued use |
| D4 | *Most* components have serious damage, *a majority* have obvious cracks | *Most* components partially destroyed | The damage is difficult to repair |
| D5 | *The majority* components have serious damage, the building structure is close to collapse or already collapsed | Non-structural components are *commonly* destroyed | To repair the building back to normal is impossible |

5    Notes about qualifiers: "very few": <10%; "a few": 10%-50%; "most": 50%-70%; "majority": 70%-90%; "commonly": >90%.

**Table 3: Divisions of seismic design level for Chinese buildings (Lin and Xie et al., 2010)**

| Seismic Resistance Design Level (PGA) | Construction Age | | | |
|---|---|---|---|---|
| | before 1978 | 1979-1989 | 1989-2001 | After 2001 |
| IX (0.4g) | pre-code | moderate-code | high-code | high-code |
| VIII (0.3g) | pre-code | moderate-code | moderate-code | high-code |
| VIII (0.2g) | pre-code | low-code | moderate-code | high-code |
| VII (0.15g) | pre-code | low-code | low-code | moderate-code |
| VII (0.10g) | pre-code | pre-code | low-code | low-code |
| VI (0.05g) | pre-code | pre-code | pre-code | low-code |

10       **Table 4. The fragility curve fitting parameters derived for empirical and analytical data.**

| data_source | build_type | fort _level | damage_state | $\mu_{LS}$ | $\sigma_{LS}$ | R-square |
|---|---|---|---|---|---|---|
| Empirical | masonry | A | LS1 | 6.926 | 1.539 | 0.99 |
| | | | LS2 | 8.418 | 1.378 | 1 |
| | | | LS3 | 9.412 | 1.189 | 1 |
| | | | LS4 | 10.57 | 1.298 | 1 |
| | | B | LS1 | 7.658 | 1.393 | 0.98 |




| | | | | | | |
|---|---|---|---|---|---|---|
| | | | LS2 | 9.283 | 1.298 | 0.99 |
| | | | LS3 | 10.43 | 1.505 | 0.99 |
| | | | LS4 | 11.59 | 1.553 | 1 |
| | RC | A | LS1 | 7.779 | 1.304 | 1 |
| | | | LS2 | 9.057 | 0.9367 | 1 |
| | | | LS3 | 9.893 | 0.9269 | 1 |
| | | | LS4 | 10.95 | 0.9626 | 1 |
| | | B | LS1 | 8.135 | 1.191 | 1 |
| | | | LS2 | 9.511 | 1.067 | 1 |
| | | | LS3 | 10.54 | 0.8831 | 1 |
| | | | LS4 | 11.77 | 1.075 | 1 |
| Analytical | masonry | A | LS1 | 0.1732 | 0.7512 | 1 |
| | | | LS2 | 0.33 | 0.7512 | 1 |
| | | | LS3 | 0.5862 | 0.6383 | 0.99 |
| | | | LS4 | 0.9416 | 0.4983 | 0.97 |
| | | B | LS1 | 0.3499 | 0.7573 | 1 |
| | | | LS2 | 0.6743 | 0.8101 | 1 |
| | | | LS3 | 1.281 | 0.8125 | 1 |
| | | | LS4 | 2.595 | 0.8581 | 0.99 |
| | RC | A | LS1 | 0.223 | 0.6615 | **0.80** |
| | | | LS2 | 0.353 | 0.7699 | **0.77** |
| | | | LS3 | 0.694 | 0.6111 | **0.73** |
| | | | LS4 | 1.404 | 0.4818 | 0.98 |
| | | B | LS1 | 0.315 | 0.539 | 0.99 |
| | | | LS2 | 0.46 | 0.5269 | 0.99 |
| | | | LS3 | 0.811 | 0.346 | 0.95 |
| | | | LS4 | 1.374 | 0.1763 | 0.91 |

*Note: "**fort_level**" **A** & **B** represent the pre/low/moderate-code and high-code seismic resistance level, respectively; "**damage_state**" LS1, LS2, LS3, LS4 represent the four damage limit states: slight, moderate, serious-, collapse, respectively; "$\mu_{LS}$" and "$\sigma_{LS}$" are the regression parameters between intensity or PGA and the corresponding fragilities of each damage limit state; **R-square** indicates the fitness quality of the regressed fragility curve.

5                     **Table 5: The mean PGA derived only based on fragilities of "masonry_A".**

| intensity | VI | VII | VIII | IX | X |
|---|---|---|---|---|---|
| PGA(g) | 0.1 | 0.16 | 0.3 | 0.48 | 0.78 |

**Table 6: The PGA ranges derived from robust intensity-PGA relations.**

| intensity | VI | VII | VIII | IX | X |
|---|---|---|---|---|---|
| PGA(g) | 0.06-0.14 | 0.12-0.25 | 0.21-0.43 | 0.36-0.73 | 0.58-1.25 |

**Table 7: The recommended intensity-PGA relationship in China (GB17742-2008/1980)**

| intensity | | VI | VII | VIII | IX | X |
|---|---|---|---|---|---|---|
| PGA (g) | mean | 0.06 | 0.13 | 0.25 | 0.5 | 0.1 |
| | range | 0.05-0.09 | 0.09-0.18 | 0.18-0.35 | 0.35-0.7 | 0.7-1.4 |

**Table 8: The latest intensity-PGA relation study in mainland China (Ding, 2017)**

| intensity | | VI | VII | VIII | IX |
|---|---|---|---|---|---|
| PGA (g) | mean | 0.09 | 0.16 | 0.3 | 0.55 |
| | range | 0.06-0.12 | 0.09-0.18 | 0.22-0.41 | 0.41-0.75 |



**Table B1: Statistics of fragility datasets for each limit state of each building type that collected from both empirical observation and analytical studies.**

| data source | build_type | intensity/ PGA(g) | original fragility number | fragility number after removing outliers | | | | median value of each fragility dataset with truncated exceed. prob. ≥ 1% | | | | standard deviation of each truncated fragility dataset with median exceed. prob. ≥ 1% | | | |
|---|---|---|---|---|---|---|---|---|---|---|---|---|---|---|---|
| | | | | LS1 | LS2 | LS3 | LS4 | LS1 | LS2 | LS3 | LS4 | LS1 | LS2 | LS3 | LS4 |
| empirical | masonry_A | 6 | 29 | 28 | 28 | 28 | 28 | 0.30 | 0.06 | 0.01 | | 0.13 | 0.04 | 0.01 | |
| | | 7 | 29 | 29 | 26 | 26 | 27 | 0.47 | 0.14 | 0.04 | | 0.21 | 0.08 | 0.04 | |
| | | 8 | 29 | 29 | 29 | 25 | 26 | 0.78 | 0.40 | 0.11 | 0.03 | 0.20 | 0.27 | 0.07 | 0.02 |
| | | 9 | 28 | 28 | 28 | 28 | 25 | 0.91 | 0.64 | 0.36 | 0.11 | 0.14 | 0.27 | 0.27 | 0.06 |
| | | 10 | 28 | 27 | 26 | 28 | 28 | 0.99 | 0.90 | 0.69 | 0.33 | 0.05 | 0.15 | 0.26 | 0.22 |
| | masonry_B | 6 | 21 | 21 | 21 | 21 | 21 | 0.15 | 0.02 | | | 0.09 | 0.02 | | |
| | | 7 | 21 | 21 | 20 | 18 | 18 | 0.26 | 0.08 | 0.02 | | 0.21 | 0.10 | 0.03 | |
| | | 8 | 21 | 21 | 21 | 21 | 18 | 0.66 | 0.17 | 0.07 | 0.01 | 0.30 | 0.24 | 0.16 | 0.01 |
| | | 9 | 20 | 20 | 20 | 20 | 17 | 0.79 | 0.37 | 0.15 | 0.05 | 0.29 | 0.30 | 0.25 | 0.03 |
| | | 10 | 20 | 20 | 20 | 20 | 20 | 0.96 | 0.74 | 0.39 | 0.15 | 0.24 | 0.31 | 0.30 | 0.22 |
| | RC_A | 6 | 24 | 23 | 22 | 19 | 24 | 0.12 | | | | 0.07 | | | |
| | | 7 | 24 | 23 | 23 | 22 | 24 | 0.25 | 0.02 | | | 0.14 | 0.05 | | |
| | | 8 | 26 | 26 | 24 | 24 | 23 | 0.57 | 0.12 | 0.02 | | 0.19 | 0.12 | 0.06 | |
| | | 9 | 20 | 20 | 20 | 19 | 18 | 0.82 | 0.48 | 0.17 | 0.02 | 0.14 | 0.17 | 0.08 | 0.02 |
| | | 10 | 16 | 16 | 16 | 16 | 14 | 0.98 | 0.84 | 0.55 | 0.16 | 0.10 | 0.16 | 0.26 | 0.10 |
| | RC_B | 6 | 6 | 6 | 5 | 6 | 6 | 0.05 | | | | 0.05 | | | |
| | | 7 | 6 | 5 | 5 | 6 | 6 | 0.15 | 0.02 | | | 0.06 | 0.01 | | |
| | | 8 | 6 | 6 | 5 | 5 | 6 | 0.48 | 0.06 | | | 0.19 | 0.02 | | |
| | | 9 | 5 | 5 | 5 | 5 | 5 | 0.75 | 0.33 | 0.04 | | 0.20 | 0.18 | 0.11 | |
| | | 10 | 5 | 5 | 5 | 5 | 5 | 0.95 | 0.67 | 0.27 | 0.05 | 0.15 | 0.25 | 0.29 | 0.11 |
| analytical | masonry_A | 0.1 | 6 | 6 | 6 | 5 | 6 | 0.22 | 0.06 | 0.02 | | 0.14 | 0.06 | 0.01 | |
| | | 0.2 | 6 | 6 | 6 | 6 | 6 | 0.60 | 0.25 | 0.08 | 0.02 | 0.15 | 0.13 | 0.06 | 0.01 |
| | | 0.3 | 6 | 6 | 6 | 6 | 6 | 0.77 | 0.47 | 0.18 | 0.05 | 0.10 | 0.14 | 0.10 | 0.03 |
| | | 0.4 | 6 | 6 | 6 | 6 | 6 | 0.86 | 0.60 | 0.29 | 0.09 | 0.07 | 0.14 | 0.14 | 0.07 |
| | | 0.5 | 6 | 6 | 6 | 6 | 6 | 0.92 | 0.70 | 0.39 | 0.14 | 0.05 | 0.12 | 0.16 | 0.10 |
| | | 0.6 | 6 | 6 | 6 | 6 | 6 | 0.95 | 0.77 | 0.50 | 0.20 | 0.04 | 0.10 | 0.16 | 0.13 |
| | | 0.7 | 6 | 6 | 6 | 6 | 6 | 0.97 | 0.84 | 0.59 | 0.27 | 0.03 | 0.08 | 0.15 | 0.15 |
| | | 0.8 | 6 | 6 | 6 | 6 | 6 | 0.98 | 0.88 | 0.66 | 0.34 | 0.02 | 0.07 | 0.15 | 0.17 |
| | | 0.9 | 6 | 6 | 6 | 6 | 6 | 0.99 | 0.91 | 0.73 | 0.41 | 0.02 | 0.06 | 0.15 | 0.18 |
| | | 1 | 6 | 6 | 6 | 6 | 6 | 0.99 | 0.94 | 0.78 | 0.47 | 0.02 | 0.05 | 0.14 | 0.19 |
| | | 1.1 | 2 | 2 | 2 | 2 | 2 | 1.00 | 0.97 | 0.91 | 0.70 | 0.00 | 0.03 | 0.08 | 0.18 |
| | | 1.2 | 2 | 2 | 2 | 2 | 2 | 1.00 | 0.98 | 0.93 | 0.74 | 0.00 | 0.03 | 0.05 | 0.16 |
| | masonry_B | 0.1 | 6 | 6 | 6 | 6 | 6 | 0.04 | 0.02 | | | 0.05 | 0.01 | | |
| | | 0.2 | 6 | 6 | 6 | 6 | 5 | 0.21 | 0.05 | 0.01 | | 0.14 | 0.05 | 0.01 | |
| | | 0.3 | 6 | 6 | 6 | 6 | 5 | 0.43 | 0.14 | 0.04 | | 0.19 | 0.09 | 0.02 | |
| | | 0.4 | 6 | 6 | 6 | 6 | 6 | 0.59 | 0.25 | 0.08 | 0.01 | 0.21 | 0.14 | 0.05 | 0.01 |
| | | 0.5 | 6 | 6 | 6 | 6 | 6 | 0.69 | 0.37 | 0.13 | 0.03 | 0.20 | 0.17 | 0.07 | 0.02 |
| | | 0.6 | 6 | 6 | 6 | 6 | 6 | 0.76 | 0.45 | 0.18 | 0.05 | 0.17 | 0.18 | 0.09 | 0.03 |
| | | 0.7 | 6 | 6 | 6 | 6 | 6 | 0.81 | 0.53 | 0.22 | 0.07 | 0.14 | 0.18 | 0.11 | 0.04 |
| | | 0.8 | 6 | 5 | 6 | 6 | 6 | 0.86 | 0.59 | 0.28 | 0.09 | 0.06 | 0.17 | 0.13 | 0.06 |
| | | 0.9 | 6 | 5 | 6 | 6 | 6 | 0.89 | 0.65 | 0.33 | 0.11 | 0.05 | 0.16 | 0.14 | 0.08 |
| | | 1 | 6 | 5 | 6 | 6 | 6 | 0.91 | 0.70 | 0.39 | 0.13 | 0.04 | 0.15 | 0.15 | 0.10 |
| | | 1.1 | 3 | 3 | 3 | 3 | 3 | 0.93 | 0.70 | 0.42 | 0.15 | 0.09 | 0.20 | 0.23 | 0.17 |
| | | 1.2 | 3 | 3 | 3 | 3 | 3 | 0.95 | 0.75 | 0.48 | 0.19 | 0.08 | 0.19 | 0.24 | 0.19 |
| | RC_A | 0.1 | 20 | 18 | 18 | 20 | 17 | 0.07 | | | | 0.07 | | | |
| | | 0.2 | 20 | 20 | 18 | 19 | 20 | 0.42 | 0.13 | 0.01 | | 0.32 | 0.12 | 0.03 | |
| | | 0.3 | 22 | 22 | 22 | 21 | 20 | 0.72 | 0.45 | 0.05 | | 0.29 | 0.35 | 0.09 | |
| | | 0.4 | 20 | 20 | 20 | 20 | 18 | 0.78 | 0.48 | 0.10 | 0.02 | 0.26 | 0.36 | 0.21 | 0.02 |
| | | 0.5 | 13 | 12 | 13 | 13 | 11 | 0.96 | 0.89 | 0.34 | 0.03 | 0.09 | 0.25 | 0.26 | 0.04 |
| | | 0.6 | 22 | 22 | 22 | 22 | 19 | 0.93 | 0.82 | 0.33 | 0.05 | 0.30 | 0.41 | 0.32 | 0.05 |
| | | 0.7 | 11 | 11 | 11 | 11 | 10 | 0.99 | 0.96 | 0.77 | 0.06 | 0.08 | 0.22 | 0.33 | 0.06 |
| | | 0.8 | 17 | 17 | 17 | 17 | 15 | 0.88 | 0.64 | 0.37 | 0.15 | 0.38 | 0.36 | 0.37 | 0.08 |
| | | 0.9 | 12 | 11 | 12 | 12 | 11 | 1.00 | 0.99 | 0.92 | 0.14 | 0.03 | 0.16 | 0.30 | 0.11 |
| | | 1 | 16 | 16 | 16 | 16 | 15 | 0.91 | 0.70 | 0.41 | 0.25 | 0.49 | 0.38 | 0.34 | 0.13 |
| | | 1.1 | 5 | 5 | 5 | 5 | 5 | 1.00 | 0.99 | 0.99 | 0.29 | 0.00 | 0.01 | 0.03 | 0.31 |
| | | 1.2 | 14 | 14 | 14 | 14 | 14 | 0.61 | 0.68 | 0.67 | 0.39 | 0.47 | 0.43 | 0.34 | 0.27 |
| | RC_B | 0.1 | 9 | 8 | 9 | 9 | 9 | 0.02 | | | | 0.02 | | | |
| | | 0.2 | 9 | 8 | 7 | 9 | 9 | 0.18 | 0.04 | | | 0.28 | 0.02 | | |
| | | 0.3 | 11 | 11 | 11 | 10 | 11 | 0.50 | 0.22 | | | 0.35 | 0.32 | | |
| | | 0.4 | 9 | 9 | 9 | 8 | 9 | 0.65 | 0.37 | 0.04 | | 0.25 | 0.33 | 0.04 | |
| | | 0.5 | 9 | 9 | 9 | 8 | 8 | 0.79 | 0.57 | 0.08 | | 0.21 | 0.31 | 0.08 | |
| | | 0.6 | 11 | 11 | 11 | 10 | 10 | 0.93 | 0.75 | 0.20 | 0.02 | 0.18 | 0.30 | 0.10 | 0.01 |
| | | 0.7 | 9 | 9 | 9 | 9 | 8 | 0.93 | 0.81 | 0.37 | 0.03 | 0.15 | 0.29 | 0.22 | 0.02 |
| | | 0.8 | 8 | 8 | 8 | 8 | 7 | 0.91 | 0.79 | 0.45 | 0.03 | 0.12 | 0.26 | 0.29 | 0.02 |
| | | 0.9 | 10 | 10 | 10 | 10 | 9 | 0.99 | 0.93 | 0.68 | 0.03 | 0.12 | 0.25 | 0.33 | 0.03 |
| | | 1 | 7 | 7 | 7 | 7 | 7 | 0.94 | 0.83 | 0.52 | 0.05 | 0.09 | 0.22 | 0.33 | 0.04 |
| | | 1.1 | 4 | 4 | 4 | 4 | 4 | 1.00 | 0.97 | 0.89 | 0.08 | 0.02 | 0.05 | 0.19 | 0.07 |
| | | 1.2 | 6 | 5 | 5 | 5 | 6 | 1.00 | 0.99 | 0.98 | 0.23 | 0.00 | 0.01 | 0.06 | 0.12 |



Note: "origin fragility number" refers to the number of original fragilities collected for each damage limit state of each building type from previous studies; "fragility number after removing outliers" refers to the remaining fragilities after removing outliers using box-plot check method. Only intensity and PGA values with truncated exceedance probability ≥1% for each damage limit state of each building type are given, since higher damage states can appear only for higher intensities or PGA values.

5    **Table B2: Chinese Official Seismic Intensity Scale: GB17742-2008**

| Macro Intensity | Senses by people on the ground | Degree of building damage | | | Other damages | Horizontal motion on the ground | |
|---|---|---|---|---|---|---|---|
| | | Building type | Damages | Mean damage index | | Peak acceleration(m/s²) | Peak speed (m/s) |
| I | Insensible | | | | | | |
| II | Sensible by very few still indoor people | | | | | | |
| III | Sensible by a few still indoor people | | Slight rattle of doors and windows | | Slight swing of suspended objects | | |
| IV | Sensible by most people indoors, a few people outdoors; a few wake up from sleep | | Rattle of doors and windows | | Obvious swing of suspended objects; vessels rattle | | |
| V | Commonly sensible by people indoors, sensible by most people outdoors; most wake up from sleep | | Noise from vibration of doors, windows, and building frames; falling of dusts, small cracks in plasters, falling of some roof tiles, bricks falling from a few roof-top chimneys | | Rocking or flipping of unstable objects | 0.31 (0.22-0.44) | 0.03 (0.02-0.04) |
| VI | Most unable to stand stably, a few scared to running outdoors | A | A few have D3 damage | 0-0.11 | Cracks in river banks and soft soil; occasional burst of sand and water from saturated sand layers; cracks on some standalone chimneys | 0.63 (0.45-0.89) | 0.06 (0.05-0.09) |
| | | B | Very few have D3 damage, a few have D2 damage, most are intact | | | | |
| | | C | Very few have D2 damage, the majority are intact | 0-0.08 | | | |
| VII | Majority scared to running outdoors, sensible by bicycle riders and people in moving motor vehicles | A | A few have D4 and/or D5 damage, most have D3 and/or D2 damage | 0.09-0.31 | Collapse of river banks; frequent burst of sand and water from saturated sand layers; many cracks in soft soils; moderate destruction of most standalone chimneys | 1.25 (0.90-1.77) | 0.13 (0.10-0.18) |
| | | B | A few have D3 damage, most have D2 and/or D1 damage | | | | |
| | | C | A few have D3 and/or D2, most are intact | 0.07-0.22 | | | |
| VIII | Most swing about, difficult to walk | A | A few have D5 damage, most have D4 and/or D3 damage | 0.29- 0.51 | Cracks appear in hard dry soils; severe destruction of most standalone chimneys; tree tops break; death of people and cattle caused by building destruction | 2.50 (1.78-3.53) | 0.25 (0.19-0.35) |
| | | B | Very few have D5 damage, most have D3 and/or D2 damage | | | | |
| | | C | A few have D4 and/or D3 damage, most have D2 damage | 0.2-0.4 | | | |
| IX | Moving people fall | A | Most have D4 and/or D5 damage | | Many cracks in hard dry soils; possible cracks and dislocations in bedrock; frequent landslides and collapses; collapse of many standalone | 5.00 (3.54-7.07) | 0.50 (0.36-0.71) |
| | | B | A few have D5 damage, most have D4 and/or D3 damage | 0.49-0.71 | | | |
| | | C | A few have D5 and/or D4 damage, | 0.38-0.6 | | | |




| | | | most have D3 and/or D2 damage | | chimneys | | |
|---|---|---|---|---|---|---|---|
| **X** | Bicycle riders may fall; people in unstable state may fall away; sense of being thrown up | A | Commonly have D5 damage | 0.69-0.91 | Cracks in bedrock and earthquake fractures; destruction of bridge arches founded in bedrock; foundation damage or collapse of most standalone chimneys | 10.00 (7.08-14.14) | 1.00 (0.72-1.41) |
| | | B | The majority have D5 damage | | | | |
| | | C | Most have D5 and/or D4 damage | 0.58-0.8 | | | |
| **XI** | | A | Commonly have D5 damage | 0.89–1.0 | Earthquake fractures extend a long way; many bedrock cracks and landslides | | |
| | | B | | | | | |
| | | C | | 0.78-1.0 | | | |
| **XII** | | A | Almost all have D5 damage | 1.0 | Drastic change in landscape, mountains, and rivers | | |
| | | B | | | | | |
| | | C | | | | | |

Notes about Qualifiers: "very few": <10%; "few": 10% - 50%; "most": 50% - 70%; "majority": 70% - 90%; "commonly": >90%.