# Peer review of "State of the art of fragility analysis for major building types in China with implications for intensity-PGA relationships"

_Natural Hazards and Earth System Sciences, 2018_

## Referee Comment (RC1) · Anonymous Referee #1 · 22 Oct 2018

The authors of this paper attempted to establish the Chinese fragilities as functions of macro-seismic intensity and PGA by reviewing the past research results. An intensity-PGA relationship is also developed in this study using the fragilities function as a bridge. However, as mentioned in the following specific comments, there are many flaws existed in the utilized methods. The authors misunderstood the concept of macroseismic intensity and are not quite familiar with the intensity evaluation work in China. The corresponding conclusion is not convincing and has limited value in engineering practice. What's more, the studied data and most of the Chinsese references in this article are collected from just one Chinese literature (Ding, 2016; Doctoral thesis.). It is not proper that the authors claimed that they "scrutinize 69 papers. . .".

[Figure]

(1) The macro-seismic intensity is usually evaluated considering the damage situation over the whole city or area, which is influenced by a lot of factors including the local economy level, population and the building types. Same magnitude earthquake events may cause totally different macro-seismic intensity levels in different cities or regions. In China, intensity evaluation work is a complex task requiring experts to adjust to local policies and local economic development levels. Many subjective factors and government policies will significantly affect the final result. However, the fragilities functions are usually used for the vulnerability description of single building or one type of building. So it makes no sense that the authors try to establish some relationship between the macroseismic intensity and the empirical fragility curves for one or two types of buildings in China. Besides that, China had experienced significant economy boost from 1975 to 2014. Therefore the inherent implications of corresponding macroseismic intensity are not the same for earthquakes over such a long time period.

(2) A more detailed earthquake information and post-survey data should be provided for the computation of the following empirical fragilities curves. (Page 5:Line 10-40) The data download link provided by author is not valid either(Page 5:Line 27) . Please update the website link and keep it accessible which is quite important for your article. Followings are some key questions we think are ignored by the authors: 1. (Page 5:Line 20-25) How to classify the buildings into different levels and extract the corresponding damage data just from the descriptions in the reference articles? The economy and development level of different provinces in China are quite unbalanced especially considering the long time period (over 40 years). How to recognize the "building construction age and corresponding code level" for a specific earthquake event? 2. (Page 5:Line 15-20) As mentioned in the paper, Stone as well as chuandou-timber structures are typical building types in mountainous area of Tibetan, Qinghai and Sichuan with frequent earthquake as indicated in the Table 5-2 in the Ding (2016). Please explain why this kind of buildings are not included and discussed in the paper. They all contribute to the final macroseismic intensity evaluation results. 3. (Page 5:Line 17) Please explain why the authors did not use the Type A, B, C building as defined in the Chinese seismic intensity standards (GB17742-2008). What survey data could support the point of "two most widely distributed building types in China: masonry and RC buildings" ? 4. (Page 5:Line 10) Please provide detailed 112 earthquake events information used in this article. After careful references check by reviewer, some of the damage situations data or ratios in the reference articles are the roughly estimation by different authors themselves. The criterions and results are actually quite subjective and different even for the same earthquake event. The number of destructive earthquake events (112) used in this article is identical with the data mentioned in the Ding's Graduate Thesis (2016). The author should consider doing some further reference proofread work.

(3) Using the fragilities function as the bridge to derive the relationship between the macroseismic intensity and PGA neglect the uncertainties in many aspects: structural, earthquake, fragility computation methods and the data source mentioned in previous comments. Just by removing the outlier and doing some variance analysis are not enough to solve the problem of uncertainty transmission. What's more, considering that the referenced data are basically similar with the Ding (2016) and Ding (2017), there is no point that the authors compared the intensity-PGA relationship results with Ding.Besides that, only the results of the mean PGA-intensity relationship derived based on fragilities of "masonry_A" is provided in Table.5. The results of the "masonry_B" and "RC_A/B" are missing due to unknown reason.

Technical Corrections: Page 17. The Fig.1 and related content in article is not necessary. The mentioned calculation methods of analytical fragility curves and damage probability matrices are actually not carried out in this article. Page 31. Line 10. The PGA range is not the same with the table in reference Ding, 2017. Page 5. Line 15. "chuandou-timber" is not properly translated.

---

## Referee Comment (RC2) · Anonymous Referee #2 · 25 Oct 2018

The purpose of the paper is to develop fragility curves, for Chinese buildings classified in two main categories (masonry, RC) and two sub classes (A and B based on their seismic resistance level), using results from >80 papers reporting damage data from many Chinese earthquakes. Both empirical and analytical methods were applied to this regard. Empirical and analytical fragility curves for the two building typologies are derived based on median values. In addition the authors proposed relationships between PGA and macroseismic intensity IM using the derived fragility curves by eliminating the fragility values from the fragility–intensity and from the fragility–PGA relations.

Knowing that there are numerous if not huge uncertainties involved in all parts of their work the authors tried to make reasonable hypothesis in order to tackle the problem and derive some useful results in the Chinese context. For example they found by their analyses that reasonable results should emerge if the building types used for analytic calculations and those used in the empirical field studies are close enough, which however is not so obvious. All their work is based on Chinese data and China's official standards proposed by CEA regarding the structure typologies, intensity scales and damage states.

The paper is well written and very interesting as it synthesizes a great number of data form Chinese earthquakes and damages to buildings. All figures and tables are useful for understanding the work done.

General comments:
1. The state of the art chapter is not as complete as it should be. Important references from the international literature are missing, namely the work in GEM, PAGER, SYNER-G, PERPETUATE etc.
2. The methodologies applied in the different steps are clearly described and the reviewer has no major comment on that except that to his opinion the various uncertainties are treated, probably inevitably, in a quite simplified way. A comment on that should be useful.
3. The classification of the buildings in only two categories and two sub classes is an over-simplification, probably a reasonable one, for sub or underdeveloped countries, but maybe not for China any more. To the reviewer's opinion if the results of this interesting and useful work, mainly considering the huge efforts made to collect and synthesize all these data, will be generalized for any building type in China, and furthermore used for risk analysis, the final outcome will be heavily biased.
4. The accuracy of the results (fragility curves) depicted in Figures 7 (empirical) and 8 (analytical) are to the reviewer' opinion "too optimistic". The derived fragilities seem to be very low for these intensity levels, either in terms of IM or PGA and in particular for masonry structures (A or B). There are many reasons for that depending on the scatter of the data but also to the method used in particular regarding the treatment of uncertainties. The authors should compare their curves with other curves from the international literature (i.e. GEM, PAGER, SYNER-G etc). In any case they should comment on that important point.
5. According to the authors IM-PGA empirical expressions are generally region-dependent and have large scatter. This is not entirely correct. If region-dependency should mean soil conditions dependency as well, then this should be probably partially fine; but region-dependence is a much broader definition (i.e. spatial variability of ground motions etc) and to the reviewer's opinion this simplification is a certain source of huge uncertainties. PGA values are strongly dependent on site and local soil conditions. Furthermore the typology of buildings and their seismic quality in terms of seismic resistance is another crucial parameter, which again is practically "crushed" and downgraded in the regression analysis. This is obvious in the results where the difference between the different approaches is very

small. In few words the reviewer is very sceptical to the use of IM-PGA relationships in earthquake engineering and risk analysis in particular. Saying that the criticism is not made on the methodology and tools applied but on the philosophy (i.e. principles) of this methodology and the accuracy of the wished outcome.

6. To the reviewer's opinion if the results of the present work i.e. the IM-PGA tables, should be used as recommended values for IM-PGA ranges in China, it should be clearly stated that this is just for preliminary evaluations and the scatter may be very important.

Minor comment:
In table 7 (Recommended intensity-PGA relationship in China (GB17742-2008/1980)) there is an obvious error in the suggested value for Intensity X.

---

## Referee Comment (RC3) · Anonymous Referee #3 · 28 Oct 2018

The authors of the present paper claim that evaluated 69 papers from Chinese literature that document observations of seismic events occurred in densely populated areas in China over the past four decades. They provide empirical fragilities for 4 building types. Also, they provide fragilities as functions of PGA using analytical methods. The authors say that they "scrutinize 69 papers". However the 18 of them are "thesis". Is this acceptable? According to the reviewers knowledge the macroseismic intensity is influenced by many factors (e.g.: population, economy level etc.). So earthquakes of the same magnitude may cause different macroseismic intensity levels in various cities. Moreover the time period is too long. The damage stage depends not only on the type of the building but on the kind of the construction and the construction type has

changed during these years. A list of the ground motions used for this study is required. The link on page 5, line 27 does not work. Many references are not adequately written (e.g.: the journal is missing, or the pages are missing etc.)

---

## Author Comment (AC1) · 23 Nov 2018

Quotation of the general comment: "The authors misunderstood the concept of macroseismic intensity and are not quite familiar with the intensity evaluation work in China."

Reply: It's a pity that we have to admit our lack of experience in conducting actual intensity evaluation work in China. But for conducting literature review of build fragility related research, we do spare essential efforts to search and check publications illustrating these standard practices as regulated in GB/T 17742-2008 (The Chinese seismic intensity scale) and GB/T 18208.4-2011 (Post-earthquake field works—Part4: Assessment of direct loss), to make sure that our understanding of the post-earthquake intensity evaluation process is not biased.

If our understanding is not biased, the key processes in post-earthquake field investigation and macroseismic intensity determination currently in China mainly follow the method firstly raised up by researcher Hu Yuxian during the investigation work of Tonghai earthquake in the 1970s (Hu, 1988). In this method, the key ideology of "average damage index" is introduced. That means, in each field survey unit (village/town/street), the number of different types of buildings in different damage states are firstly investigated; median damage index of five damage states D5, D4, D3, D2, D1 as defined in GB17742-2008 are used in later on calculation, such as 0.93, 0.70, 0.43, 0.20, 0.05 for these five damage states; for each building type in each field survey unit, the corresponding average damage index is derived by summarizing the products of percentage of building in each damage state and its damage index; generally there should be one or two standard building types defined in advance (such as Type A, B, C in GB/T 17742-2008), based on which the empirical relationship between macroseismic intensity and building damage index was established in advance from previously historical records; then the average damage index of other surveyed building types can be further scaled to the standard building type's damage index; finally the overall average building damage index for each survey unit can be calculated by summarizing the products of each building type's average damage indexes and its occupancy ratio in the survey unit; once the average damage index for each survey unit is determined, the corresponding macroseismic intensity can be easily derived from the empirical relationship between macroseismic intensity and building damage index derived based on the standard building type.

Furthermore, the authors' knowledge in other countries' intensity evaluation work help us to keep in mind at the very beginning that intensity evaluation work is based on the damage investigation of groups of buildings in earthquake affected area and is involved with various subjective expert judgements.

Quotation of the general comment: "The corresponding conclusion is not convincing and has limited value in engineering practice."

Reply: For detailed engineering practice, we admit our results are still too general and have limited application value, but that's also not our main focus. Instead, our main focus by doing such a literature review work is to (1) have an overall picture of the fragilities given by various experts and researchers in China for various building types, not limited to masonry and RC (2) given the large uncertainty embedded in traditional practice of deriving PGA-intensity relationship, this work is trying some "novel" idea of using "fragility" as the bridge to link PGA and intensity. Sometimes, "novel" means "wired" and is often "risky". We admit the various difficulties lying in between, but since the PGA-intensity regression relationships we derived are still comparable with that derived using traditional practice like in Ding (2017), which is based on different assumption and processing methods from us. This comparability reflects the acceptability of such a new trial.

Quotation of the general comment: "What's more, the studied data and most of the Chinese references in this article are collected from just one Chinese literature (Ding, 2016; Doctoral thesis.). It is not proper that the authors claimed that they "scrutinize 69 papers. . ."

Reply: We're afraid that we cannot agree with this claim. Thanks to the first author's advantage of having Chinese as her mother tongue, for thoroughly checking the available publications related to Chinese building fragility, we checked actually much more than 69 Chinese publications but finally among them 53 have the detailed fragilities numbers that we need. The other 16 publications are complemented after comparing with earthquakes listed in Ding (2016). The overlap of literature is inevitable, since the research topic is highly similar. But our assumption and processing of fragility are quite different from that in Ding (2016). For example, although our references used are overlapped to a certain extent, however, in Ding (2016), the collection of fragility data is earthquake based; ours is literature based, which means one literature may include comprehensive fragility information derived from more than one earthquake's field survey result.

Actually, during the digitalizing process of both the empirical and analytical fragilities from our pre-downloaded publications, the first author was quite excited to find out that the PhD work of Ding (2016) had provided the collection of building fragility data in such a detailed way. However, after closer check of the data, a big inconvenience that limits the further manipulation of them is that, the fortification information of the damaged buildings is totally unknown. This is also a big difficulty acknowledged in Ding (2016). Later on, they made such an assumption that the ratio of fortificated masonry building is consistent with the overall GDP increasing trend in China during 1978-2015 (related Chinese description: "假设1978年~2015年我国GDP的增长趋势同这一时期的经过抗震设防的砖砌体房屋在总体房屋中所占的比例是一致的"; Ding, 2016, Page92). Given the fact of uneven economic development levels between eastern and western China, as well as the fact that earthquakes occurred more often in western China than in the east, we consider this assumption is kind of too rough. Thus though the work of Ding (2016) was very nice, but regretfully we couldn't directly use those fragility data.

In comparison, during our data collection work at the very beginning, we extract all the available important supplementary information including the fortification level and age of the damaged buildings besides their type and fragility from every single literature we scrutinized.

Quotation of details in Comment (1): "The macroseismic intensity is usually evaluated considering the damage situation over the whole city or area, which is influenced by a lot of factors including the local economy level, population and the building types. Same magnitude earthquake events may cause totally different macroseismic intensity levels in different cities or regions. In China, intensity evaluation work is a complex task requiring experts to adjust to local policies and local economic development levels. Many subjective factors and government policies will significantly affect the final result."

Reply: Yes, we're deeply impressed by and totally agree with this fact.

Quotation of details in Comment (1): "However, the fragilities functions are usually used for the vulnerability description of single building or one type of building. So it makes no sense that the authors try to establish some relationship between the macroseismic intensity and the empirical fragility curves for one or two types of buildings in China."

Reply: Here our understanding or definition of "fragility function" is kind of different. In our work, we classify fragility functions into two types: empirical and analytical. Empirical fragility function is

macroseismic intensity related, and analytical fragility function is instrumental peak ground motion (PGA) or spectral acceleration (SA) related.

Empirical fragility is usually given in the form of damage probability matrices (DPM) in post-earthquake investigation, indicating the percentage of each building type in different damage states given specific intensity level. And empirical fragility function can be derived from these DPMs based on normal distribution assumption, as indicated by Eq. (1) and Eq. (3) in our manuscript.

Additionally, as mentioned in Page 2 line 15-19 in the manuscript, such empirical fragility curves are very helpful in seismic risk assessment and macroseismic intensity is more directly related the real damage of buildings and infrastructures in an earthquake. While previous different approaches and methodologies are spread across scientific journals, conference proceedings, technical reports and software manuals, hindering the creation of an integrated framework that could allow the visualization, acquisition and comparison between all the existing curves, therefore we try to do such a literature review work for building fragility analysis for major building types in China. From this point of view, we insist that the establishment of macroseismic intensity related empirical fragility curve is reasonable and necessary.

Quotation of details in Comment (1): "Besides that, China had experienced significant economy boost from 1975 to 2014. Therefore the inherent implications of corresponding macroseismic intensity are not the same for earthquakes over such a long time period."

Reply: Yes, we're deeply aware of the change of building fortification performance over the 40-year time span, therefore the available building construction fortification level and age information are embedded in the classification of buildings into masonry_A, masonry_B, RC_A, RC_B types, with reference to the criteria in Table 3 of the manuscript.

Quotation of details in Comment (2): "A more detailed earthquake information and post-survey data should be provided for the computation of the following empirical fragilities curves. (Page 5:Line 10-40) The data download link provided by author is not valid either (Page 5:Line 27) . Please update the website link and keep it accessible which is quite important for your article."

Reply: Thank you for the suggestion. We'll modify the link (sorry for this unexpected technical chaos). Luckily the data linked is actually the same as those uploaded in the online supplementary material. (https://www.nat-hazards-earth-syst-sci-discuss.net/nhess-2018-254/) But we realized only those processed data used directly for plotting the figures in the manuscript is not enough. For your further check, here we also uploaded all the four original documents related to the preliminary processing of the collected fragility data: (available at https://app.box.com/s/lwlqajpogxqlau72ravew4db7y47drtk)

(1) Filename "China Economic Loss and Vulnerability function review.xlsx":
This is our original records of fragility data extracted from the aforementioned scrutinized publications. Among those mentioned 87 publications, 71 have detailed fragilities given, the other 16 publications are either concept/method related or their data have been included in those 71 publications.
Sheet "site survey and statistic 36" includes empirical fragility data extracted from 36 publications without clear building fortification information, which will be estimated from available supplementary information;
Sheet "with fortification 16" includes empirical fragility data extracted from another 16 publications with clear building fortification information;

Sheet "analytical prediction 18" includes all the analytical fragility data extracted from 18 publications' tables or figures, among them some have clear building fortification information, others not;

Sheet "Ding Baorong 2016" is the fragility data collected by Ding (2016, doctoral thesis). Due to the high similarity in research purpose, we also digitalized these data and actually reiterated his/her work based the assumptions described in Ding (2016). This comparison will be explained later on.

(2) Filename "China Vulnerability records.xlsx":

Based on the results in Step (1), in this file the fragility data are further divided into different building groups: soil-wood, brick-wood, brick-concrete, analytical_masonry, RC, analytical_RC, industrial frame, stone-wood, Chuandou-timber, wood, stone and soil, as can be checked from each sheet with the same name. Here, 'brick-concrete' equals to masonry in our nomination. But for intensity-PGA relationship development, we have to focus only on masonry and RC, since analytical fragility data for other building types are not available.

(3) Filename "China Vulnerability analysis_plot.xlsx":

In this file, the exceeding probabilities of four damage limit states (LS1, LS2, LS3, LS4) are derived using Eq. (1) in the manuscript;

Given the main focus of this work as aforementioned and data abundance of each building type's fragility data, we finally focus on Sheets "brick-concrete", "analytical_masonry", "RC" and "analytical_RC" for further fragility curve derivation and PGA-intensity relationship exploration.

To fully use the fortification information given in each literature and make less assumption, we further add different fortification tags to empirical fragility data in Sheets "brick_concrete" and "RC", and to analytical fragility data in Sheets "analytical_masonry" and "analytical_RC";

The grouping criteria are as follows:

For empirical masonry fragility data in Sheet "brick-concrete": five tags specified in Column "Group of data" are used in differentiating the collected data: no fortification, unspecified fortification, low fortification, middle fortification and high fortification.

'no fortification' means there is no available information we can get from corresponding publication;

'unspecified fortification' means that in some publication, they mentioned the building is fortificated or unfortificated, but with no detailed fortification level information;

'low fortification' refers to buildings with VI level fortification as given in corresponding literature;

'middle fortification' refers to buildings with VII level fortification as given in corresponding literature;

'high fortification' refers to buildings with ≥VIII level fortification as given in corresponding literature.

Additionally, available building age information is used in grouping of fragility data extracted, like in Sheet "brick-concrete" from Hu & Sun (2010).

For analytical masonry fragility data in Sheet "analytical_masonry", in Column "Group of data":

'low-middle fortification' refers to buildings modelled with 0.05g~0.2g fortification as described in corresponding literature;

'high fortification' refers to buildings modelled with ≥0.3g fortification as described in corresponding literature.

For empirical RC fragility data in Sheet "steel-RC", the grouping criteria are similar to that in "brick-concrete", with slight difference in that given RC buildings are generally have better fortification performance than masonry, so in publications where building fortification information is not available, we mark it as "unspecified fortification" , as can be checked in Column "Group of data" as well.

For analytical RC fragility data in Sheet "analytical_steel", the grouping criteria are similar to that in "analytical_masonry", with slight difference that since RC buildings are generally have better fortification performance than masonry, so in publications where building fortification information is not available, we also mark it as "unspecified fortification".

Besides that, in Sheet "analytical_masonry", fragility data based both on PGA and SA are collected, but since for masonry building, only PGA related fragility data are available, so finally we only use PGA related analytical fragility data for RC buildings.

(4) Filename "China Vulnerability analysis_plot_result.xlsx":
This file is not so much different from the file in Step (3), only that we regroup the data with different fortification level assigned in Step (3) and plot the fragility distribution.

To achieve certain statistical significance of the fragility data analysis, in this step subjective judgement is necessary. Thus for brick-concrete or masonry buildings, we assign "masonry_A", "RC_A" building type to represent those with unspericifed/low/middle fortification level tag in Step (3), and "masonry_B", "RC_B" to include those with high fortification level tag in Step (3), as also described in Page 5: line 18-23 in the manuscript.

The data uploaded in the online supplementary material (or the invalid personal link) is the classified fragility data extracted from this file.

The first author is quite sorry for the inconvenience caused due to the addition of the invalid personal link in Page 5: Line 27. It was meant to provide more access to the fragility data uploaded in the online supplementary material. Later on, we'll remove this link and directly recommend readers to download the online supplementary material, which is the same content as this invalid link refers to.

Quotation of question 1 in Comment (2): "1. (Page 5: Line20-25) How to classify the buildings into different levels and extract the corresponding damage data just from the descriptions in the reference articles? The economy and development level of different provinces in China are quite unbalanced especially considering the long time period (over 40 years). How to recognize the "building construction age and corresponding code level" for a specific earthquake event? "

Reply: As explained in detail in last response, the fragility data was extracted from tables or digitalized from figures using software "GetData Graph Digitizer" when this kind of information is available. Not all publications give construction age and fortification level of the damage buildings, detailed processing steps are also explained in previous response. And the fragility data is not collected earthquake by earthquake, that's Ding (2016)'s practice; for us we have limited access to all the publications, instead we collected fragility data from each literature with available supplementary information, which is lacked in Ding (2016).

Quotation of question 2 in Comment (2): "2. (Page 5: Line 15-20) As mentioned in the paper, Stone as well as chuandou-timber structures are typical building types in mountainous area of Tibetan, Qinghai

and Sichuan with frequent earthquake as indicated in the Table 5-2 in the Ding (2016). Please explain why this kind of buildings are not included and discussed in the paper. They all contribute to the final macroseismic intensity evaluation results."

Reply: In our work, empirical fragility data is available for soil-wood, brick-wood, brick-concrete, RC, industrial frame, stone-wood, chuandou-timber, wood, stone and soil; but analytical fragility data is only available for masonry and RC. Since another focus of our work is to explore PGA-intensity relationship using fragility as conversion, that's why finally only masonry and RC data are further used.

Quotation of question 3 in Comment (2): "3. (Page 5:Line 17) Please explain why the authors did not use the Type A, B, C building as defined in the Chinese seismic intensity standards (GB17742-2008). What survey data could support the point of "two most widely distributed building types in China: masonry and RC buildings"?"

Reply: Thank you for this suggestion. We'd like to. However, since the building information is purely extracted from each literature, most of their fortification descriptions are related to intensity or PGA level, not in the form of Type A, B, C, that's why we couldn't use this type of building description.

The literature source for "two most widely distributed building types in China: masonry and RC buildings" is Sun & Chen (2009), Page 3, Para 1.2.1. The related Chinese description is "根据调查结果：我国城市 70%以上其砌体结构，15%以上是钢筋混凝土结构，其它结构形式所占比例较小…"（孙柏涛，陈洪富，地震工程与工程震动，"计及城市房屋建筑装修破坏的地震经济损失评估方法研究"，2009.）

Quotation of question 4 in Comment (2):" 4. (Page 5: Line 10) Please provide detailed 112 earthquake events information used in this article. After careful references check by reviewer, some of the damage situations data or ratios in the reference articles are the roughly estimation by different authors themselves. The criterions and results are actually quite subjective and different even for the same earthquake event. The number of destructive earthquake events (112) used in this article is identical with the data mentioned in the Ding's Graduate Thesis (2016). The author should consider doing some further reference proofread work."

Reply: The number of events "112" is indirectly derived after careful checking the literature we used with that cited in Ding (2016), to avoid missing any important reference for us or at least to make sure that the 112 earthquakes listed in Ding (2016) have all been covered by the publications we refer to. We believe this number provided by Ding (2016) is trustworthy enough. We also cited Ding (2016) in Page 2 Line 34 in the manuscript when first mentioned the number "112".

One major difference between our work is that, as also mentioned previously, in Ding (2016)'s work, the damage data is collected earthquake by earthquake; in our work, the fragility data provided in some references may be an average result from previous earthquakes ever occurred in specific area. These explanatory information can be checked from the additionally uploaded file "China Economic Loss and Vulnerability function review.xlsx", Sheet "site survey and statistic 36", Column "EQ details".

As to the quality or robustness of each of the fragility data given in various references we cited, we're also quite aware of the fact that even experts' judgement can be quite different, let alone those not so trustworthy results. That's why we try to collect as many data as possible and put them together to tell the trend. We do refer to a lot, but don't trust them all. We assume that every researcher has been rigorous in the data they provide, although that's not 100% the case. We conduct the following process

to avoid been misled by those untrustworthy data. First, in later on fragility curve derivation, we choose to use the median fragility instead of the mean fragility for each intensity or PGA level; Secondly, we believe bad data "speak" for themselves. If they deviated too much from the "appropriate" value, they'll be easily marked as "outliers" using box-plot method.

Besides that, there is another flaw in the data we collected that hasn't been pointed out by you but we think it's necessary to explain. That is the inter-citation of fragility data in different literature. Since the most reliable and authoritative sources tend to be most widely cited, we consider this to certain extent won't be a problem, since the increasing citation of those reliable data decreases their chances of being marked as "outliers" and increases their chances as "median" fragility, which in turn ensure the robustness of the fragility value to be used to derive the corresponding fragility curve.

Quotation of details in Comment (3): "Using the fragilities function as the bridge to derive the relationship between the macroseismic intensity and PGA neglect the uncertainties in many aspects: structural, earthquake, fragility computation methods and the data source mentioned in previous comments. Just by removing the outlier and doing some variance analysis are not enough to solve the problem of uncertainty transmission."

Reply: We totally agree that uncertainty is prevailing in every single step both in empirical and analytical fragility analysis, as pointed out in Page 2 line 1-13 in the manuscript.

In this regard, besides carefully classifying the originally collected fragility data and conducting box-plot analysis, we also provided the standard deviation of the overall fragility distribution for each intensity and PGA level in Appendix Figure A1-A4 and Table B1.

Further beyond that, we also spare quite a few efforts in charactering the uncertainty transmission from the fragility curve to the PGA-intensity relationship based on the data in the Appendix. To avoid the manuscript to appear to be too sparse and extensive, we didn't extend this uncertainty characterization process and only provide a reference uncertainty value of "0.3" in Page 9 line 9 for Eq. (5). However, for your further check, we put this methodology description in additionally uploaded file "Methodology of uncertainty transmission.pdf".

(This file is available at https://app.box.com/s/lwlqajpogxqlau72ravew4db7y47drtk)

Quotation of details in Comment (3): "What's more, considering that the referenced data are basically similar with the Ding (2016) and Ding (2017), there is no point that the authors compared the intensity-PGA relationship results with Ding. Besides that, only the results of the mean PGA-intensity relationship derived based on fragilities of "masonry_A" is provided in Table.5. The results of the "masonry_B" and "RC_A/B" are missing due to unknown reason."

Reply: As detailed explained before, actually the PGA data we use and that in Ding (2016) are different. For the PGA-intensity relationship development, the main difference is that in Ding (2017), PGA data are actually instrumental records collected from the same geographical range as macroseismic intensity records. This kind of traditional practice in deriving PGA-intensity relationship generally inevitably has high scatter and regional dependence (Caprio, 2015).

In our work, firstly the PGA values we used actually are not real instrumental records, but the PGA parameters used in analytical fragility studies; secondly, before fitting PGA and intensity, we try to make some classification based on the difference in building fortification performance, hopefully this kind of further classification can help narrow down the scatter in traditional practice. Finally our PGA-intensity relationship is indirectly derived by using the "fragility" as the bridge. If this kind of

"novel/risky/wired" practice of deriving PGA-intensity relationship can derive comparable results with those based on true records as in Ding (2017), this indirectly reflects the reasonability of our trial.

Table 5 in the manuscript is the PGA ranges for each intensity level derived from "masonry_A" , the consideration to show this for "masonry_A" separately is explained in Page 8;

Table 6 is the combined PGA ranges derived from "masonry_A", "masonry_B" and "RC_B". "RC_A" is not used due to the high scatter in the originally collected fragility data, which is also explained in detail in Page 8.

Quotation of details in Technical Corrections: "The Fig.1 and related content in article is not necessary."

Reply: Since Fig.1 helps to clarify generally how empirical and analytical fragility curves are derived, which is one of the main focus of this work, we think it's helpful and also necessary.

Quotation of details in Technical Corrections: "The mentioned calculation methods of analytical fragility curves and damage probability matrices are actually not carried out in this article."

Reply: Since this work is kind of literature review oriented, we think it's necessary to briefly introduce the mathematical methods used in previous literature, to make the work seems more self-consistent.

Quotation of details in Technical Corrections: "Page 31. Line 10. The PGA range is not the same with the table in reference Ding, 2017."

Reply: Thank you for such careful check. Since our collection of empirical fragility data are for intensity VI-X, that's why we didn't put the PGA ranges for intensity V from Ding (2017), not deliberately removing it. But after closer check of the PGA range in Table 8, we found out a type-in error for intensity VII, the corresponding PGA range should be 0.09-0.22. We'll rectify this in the revised version.

Reply to details in Technical Corrections: " Page 5. Line 15. "chuandou-timber" is not properly translated."

Reply: We searched published literature and find four forms of related expression: (1) masonry-infilled timber house (2) chuan-dou timber house (3) chuan-dou style wood frame (4) Dieh-Dou timber frame used in Taiwan. But still, more hints are sincerely anticipated, if chuandou-timber is not that exact description.
* * *
Given the high similarity of research objectives between our work and those in Ding (2016) and Ding (2017), though we follow different data collection and manipulation processes, for your further check, we also additionally upload our reiteration of Ding (2016)'s fragility data analysis work [filename: Ding2016_fragility_plot.xlsx] by following the assumptions made in it. (This file is available at https://app.box.com/s/lwlqajpogxqlau72ravew4db7y47drtk)

Here we take masonry building for example to show the difference in derived fragility curves:

In Ding (2016), unfortificated and fortificated buildings over the years are divided in accordance to China's growth rate per year; and finally they use the "mean" PGA to represent the fragility for each intensity level; in our reiteration work, we also additionally use their median PGAs for each intensity level and plot them together with the fragility curves we derived based on our data collection and classification process:

Legend explanation:

XDH_median: fragility curves derived using our median fragility data;

DBR_median: fragility curves derived using Ding (2016)'s median fragility data;

DBR_mean: fragility curves derived using Ding (2016)'s mean fragility data.

[Figure]

Figure 1: In "masonry_unfort", the data used are Ding (2016)'s unfortificated fragility data and our collected data with no available fortification information given;

[Figure]

Figure 2: In "masonry_A", the data used are Ding (2016)'s fortificated fragility data and our collected data assigned as "masonry_A";

[Figure]

Figure 3: In "masonry_B, the data used are Ding (2016)'s fortificated fragility data and our collected data assigned as "masonry_B";

As can be seen, for Ding (2016)'s data, the usage of median or mean fragility (which is subjective) can lead to different results. Fragility curves derived from Ding (2016)'s unfortificated and our unfortificated data seem to overlap much better than for those sfortificated data. Since these two works follow different data collection sources (Ding's earthquake oriented and ours' literature oriented), based on different assumptions in data processing, which method is better than another is yet difficult to tell. But due to the uncertainties embedded in every single step, we'll add essential limitation and reminding to the usage of the relationships we derived in the revised manuscript if possible.

**References:**

Caprio, M., Tarigan, B. and Worden, C. B.: Ground motion to intensity conversion equations (GMICEs): A global relationship and evaluation of regional dependency, Bulletin of the Seismological Society of America, 105, 1476--1490, 2015.

Ding, B.: Study on Related Quantitative Parameters of Seismic Intensity Scale, Thesis, Institute of Engineering Mechanics, China Earthquake Administration, Harbin, 195, 2016.

Ding, B., Sun, J. and Du, K.: Study on relationships between seismic intensity and peak ground acceleration, peak ground velocity, Earthquake Engineering and Engineering Dynamics, 26-36, 2017.

Hu, Y.: Earthquake Engineering, Earthquake Press, Beijing, China, 1988

Sun, B. and Chen, H.: Urban Building Loss Assessment Method Considering the Decoration Damage due to Earthquake, Journal of Earthquake Engineering and Engineering Vibration, 164-169, 2009.

---

## Author Comment (AC2) · 15 Dec 2018

Quotation of the general comment 1: "The state of the art chapter is not as complete as it should be. Important references from the international literature are missing, namely the work in GEM, PAGER, SYNER-G, PERPETUATE etc."

Response: Thank you for this very good suggestion. Although our focus is on building types and damage data within China, it's always worthwhile to conduct necessary comparison with similar projects worldwide. Therefore, we checked their manuals and projects reports carefully. Comparison details are as follows:

(a) For European PERPETUATE project, its main goal was to develop European Guidelines for evaluation and mitigation of seismic risk to cultural heritage assets, applicable in the European and other Mediterranean countries. The assessment of heritage buildings requires the assessment of both architectonic and artistic assets contained in them, which needs improvement in methods of analysis and assessment procedures rather than in intervention techniques. Besides that, a verification approach in terms of displacement rather than in terms of strength is more reliable and effective for heritage building. However, the fragility we collected in this work is mainly macroseismic intensity or PGA related. Therefore, the fragility outputs are not so comparable.

(b) For European SYNER-G project, the mainly studied building types are also masonry and RC and it has highly similar focuses as to our work, namely (1) to collect existing fragility functions (2) to identify categories for grouping buildings (3) to harmonize different intensity measures and limit states. And finally, all their fragility outputs are related to PGA, with some converted from macroseismic intensity, SA related fragility functions.

What's more, there are only two referred damage limit states in their output, namely yielding and collapse. For instance, if three limit states are considered (LS1, LS2 and LS3), the user can decide to assign LS1 to yielding and LS3 to collapse. Otherwise, he/she can also decide to assign a mean between LS1 and LS2 to yielding limit states. Hereafter, we use LS2 and LS4 to represent the "yielding" and "collapse" damage state in SYNER-G project.

It's worth to note that, SYNER-G project proposed a new modular form building taxonomy, based on difference in building resisting mechanism and material, floor/roof system, height level, code level etc., which is more expandable compared with existing building taxonomies like PAGER (tailored for worldwide structures) and RISK-UE (suited to Europe).

From our point of view, besides the epistemic and aleatory uncertainties imbedded within the standard fragility generation process itself, the conversion from intensity/SA to PGA and the simplification in damage state harmonization in SYNER-G project's fragility outputs inevitably add extra uncertainty to the final

output fragility results.

In spite of the differences in building classification, damage state harmonization between SYNER-G and our work, we plotted the fragility outputs together in Fig. 1 for masonry building type and in Fig. 2 for RC building types. Two obvious characteristics can be found in Fig. 1 and Fig. 2. Firstly, the fragility in SYNER-G project is much higher than ours, both for masonry and RC building types; Secondly, for SYNER-G RC building types, the fragility difference is very subtle for yielding damage state (LS2).

The reason for this obvious fragility difference between Chinese masonry/RC and European masonry/RC is difficult to fully examine, as aforementioned, it may due to the difference in usage of ground motion indicator (part of SYNER-G's PGA-related fragility outputs are converted from intensity, SA related fragility functions). Besides, building classification difference is difficult to accurately calibrate. Furthermore, the damage limit state harmonization in SYNER-G (only yielding and collapse damage states) makes it more difficult to compare the fragility for nominally similar building type for each damage state.

[Figure]

Figure 1: Fragility comparison between SYNER-G project outputs and our work for masonry building. In SYNER-G project, Masonry_A, Masonry_B refer to the low-rise, mid-rise building type, respectively; LS2 and LS4 specially refer to yielding state and collapse state, respectively.

[Figure]

Figure 2: Fragility comparison between SYNER-G project outputs and our work for RC buildings. In SYNER-G project, RC_A, RC_B, RC_C, RC_D refer to four RC subtypes, namely RC mid-rise with moment resisting frame (RC_A), RC mid-rise with lateral load design (RC_B), RC mid-rise with bare moment resisting frame with lateral load design (RC_C), and RC mid-rise with bare non-ductile moment resisting frame with lateral load design (RC_D); LS2 and LS4 specially refer to yielding state and collapse state respectively.

(c) For US's PAGER project, it's an automated system mainly for rapidly estimating the shaking distribution, the number of people and settlements exposed to severe shaking, as well as the range of possible fatalities and economic losses. During this process, vulnerability functions are used, which are different from the fragility functions we focus on in this work. That is, vulnerability functions can be derived directly from historic damage information, or derived indirectly from fragility function using consequence functions, which describe the probability of loss given a level of performance (e.g. collapse damage state). Therefore, direct comparison between the outputs of PAGER and our fragility functions is not straightforward.

(d) For US's HAZUS project, with the vision to provide local, state and regional officials with the tools necessary to plan and stimulate efforts to reduce risk from earthquakes and to prepare for emergency response and recovery from an earthquake, HAZUS offers a series of fragility curves for typical building types based on HAZUS taxonomy. Here, we extracted the equivalent PGA related fragility curves for four typical building types (RM1M, RM2M, C1M, C2M) from HAZUS Earthquake Technical Manual (from their Table 5.16a-d) and compare them with the fragility curves we developed for masonry in Fig. 3, Fig. 4 and for RC in Fig. 5, Fig. 6.

It's worth to note that, the present HAZUS curves represent median values of equivalent-PGA fragility curves. They are based on median values of spectral displacement of the damage state of interest and an assumed demand spectrum shape that relates spectral response to PGA. As such, median values of equivalent PGA are very sensitive to the shape assumed for the demand spectrum. The reference spectrum

represents ground shaking of a large-magnitude (i.e., M ≅ 7.0) western United States (WUS) earthquake for soil sites (e.g., Site Class D) at site-to-source distances of 15 km, or greater.

From Fig. 3, we can see that the order of fragility is basically as follows (given same PGA level, which building type is more "fragile"):

For damage state LS1, LS2, LS4, RM1M_Highcode<XDH_Masonry_A< RM1M_Modecode;

For damage state LS3, RM1M_Modecode<XDH_Masonry_A<RM1M_Lowcode;

For damage state LS1, LS2, LS3, LS4, XDH_Masonry_B< RM1M_Highcode.

[Figure]

Figure 3: Fragility comparison between HAZUS RM1M building and our work for RC buildings. In HAZUS project, "RM1M" refers to Mid-rise Reinforced Masonry Bearing Walls with Wood or Metal Deck Diaphragms; LS1, LS2, LS3, LS4 refer to slight damage, moderate damage, extensive damage and collapse damage states.

From Fig. 4, we can see that the order of fragility is basically as follows:

For damage state LS1, LS2, XDH_Masonry_A< RM2M_Highcode;

For damage state LS3, RM2M_Highcode <XDH_Masonry_A< RM2M_Modecode;

For damage state LS4, RM2M_Modecode<XDH_Masonry_A<RM2M_Lowcode;

For damage state LS1, LS2, LS3, LS4, XDH_Masonry_B< RM2M_Highcode.

[Figure]

Figure 4: Fragility comparison between HAZUS RM2M building and our work for RC buildings. In HAZUS project, "RM2M" refers to Mid-rise Reinforced Masonry Bearing Walls with Precast Concrete Diaphragms; LS1, LS2, LS3, LS4 refer to slight damage, moderate damage, extensive damage and collapse damage states.

Based on the analysis in Page 8, Line 1-17 in the manuscript, the fragility curves of LS1-LS4 of RC_A and LS4 of RC_B are not so reliable; therefore, we mainly compare the fragility curves of LS1-LS3 of RC_B with HAZUS C1M building type in Fig. 5 and C2M building type in Fig. 6.

From Fig. 5, we can see that the order of fragility is basically as follows:

For damage state LS1, LS2, XDH_RC_B<C1M_Highcode;

For damage state LS3, XDH_RC_B≈C1M_Highcode.

[Figure]

Figure 5: Fragility comparison between HAZUS C1M building and our work for RC buildings. In HAZUS project, "C1M" refers to Mid-rise Concrete Moment Frame; LS1, LS2, LS3, LS4 refer to slight damage, moderate damage, extensive damage and collapse damage states.

From Fig. 6, we can see that the order of fragility is basically as follows:

For damage state LS1, LS2, XDH_RC_B<C2M_Highcode;

For damage state LS3, XDH_RC_B≈C2M_Highcode.

[Figure]

Figure 6: Fragility comparison between HAZUS C2M building and our work for RC buildings. In HAZUS project, "C2M" refers to Mid-rise Concrete Shear Walls; LS1, LS2, LS3, LS4 refer to slight damage, moderate damage, extensive damage and collapse damage states.

To summarize, due to the difference in building classification and seismic resistance level harmonization between HAZUS and our work (as we only have level A and B), it's difficult to whose "masonry" or "RC" is more "fragile" than another.

(e) For the currently undergoing global-scale GEM project, it's involved in outputs from 3 European programs: SHARE, SYNER-G and NERA. SHARE focuses on seismic hazard harmonization in Europe and covers all of Europe and the Maghreb countries, and the hazard model is developed with the OpenQuake Engine. SYNER-G partners are developing a unified methodology and tools for systemic vulnerability assessment in Europe. NERA focuses on creation of a European research infrastructure for risk assessment and mitigation.

Besides the fragility outputs of SYNER-G project, GEM online fragility database also integrates those many fragility curves generated by HAZUS. Therefore, we don't repeat the comparison of these data. For mainland China, the fragility curves integrated in GEM database is solely from Tang et al. (2011), only for RC building and related to SA. Therefore, to avoid uncertainty introduced from converting SA to PGA, here we don't list and compare the fragility either.

Quotation of the general comment 2: "The methodologies applied in the different steps are clearly described and the reviewer has no major comment on that except that to his opinion the various uncertainties are treated, probably inevitably, in a quite simplified way. A comment on that should be useful."

Response: Thank you for specially pointing this out. We do spare quite a few efforts in charactering the uncertainty transmission from the fragility curve to the PGA-intensity relationship based on the data in the Appendix. To avoid the manuscript to appear to be too sparse and extensive, we didn't extend this uncertainty characterization process and only provide a reference uncertainty value of "0.3" in Page 9 line 9 for Eq. (5). However, for your further check, we put this methodology description in additionally uploaded file "Methodology of uncertainty transmission.pdf".
(This file is available at https://app.box.com/s/lwlqajpogxqlau72ravew4db7y47drtk)

Quotation of the general comment 3: "The classification of the buildings in only two categories and two sub classes is an over-simplification, probably a reasonable one, for sub or underdeveloped countries, but maybe not for China any more. To the reviewer's opinion if the results of this interesting and useful work, mainly considering the huge efforts made to collect and synthesize all these data, will be generalized for any building type in China, and furthermore used for risk analysis, the final outcome will be heavily biased."

**Response**: In our initial fragility data collection work, we actually collected fragility data for more buildings types, including soil-wood, brick-wood, brick-concrete, RC, industrial frame, stone-wood, chuandou-timber, wood, stone and soil; but analytical fragility data is only available for masonry and RC. Since another focus of our work is to explore PGA-intensity relationship using fragility as conversion, that's why finally only masonry and RC data are further used. Due to the uncertainty in the synthetization process, we agree that if more building types are to be used, the final outputs can be quite different. Here the usage of fragility as the bridge to derive intensity-PGA relationship, instead of aiming to provide a precise relationship, is more targeted at presenting a new trail to regress intensity-PGA relation.

**Quotation of the general comment 4**: "The accuracy of the results (fragility curves) depicted in Figures 7 (empirical) and 8 (analytical) are to the reviewer' opinion "too optimistic". The derived fragilities seem to be very low for these intensity levels, either in terms of IM or PGA and in particular for masonry structures (A or B). There are many reasons for that depending on the scatter of the data but also to the method used in particular regarding the treatment of uncertainties. The authors should compare their curves with other curves from the international literature (i.e. GEM, PAGER, SYNER-G etc). In any case they should comment on that important point."

**Response**: Thank you for this suggestion. We checked the fragility database of those several projects and detailed comparison can be referred to first Response to comment 1.

**Quotation of the general comment 5**: "According to the authors IM-PGA empirical expressions are generally region-dependent and have large scatter. This is not entirely correct. If region-dependency should mean soil conditions dependency as well, then this should be probably partially fine; but region-dependence is a much broader definition (i.e. spatial variability of ground motions etc.) and to the reviewer's opinion this simplification is a certain source of huge uncertainties. PGA values are strongly dependent on site and local soil conditions. Furthermore,the typology of buildings and their seismic quality in terms of seismic resistance is another crucial parameter, which again is practically "crushed" and downgraded in the regression analysis This is obvious in the results where the difference between the different approaches is very small. In few words the reviewer is very sceptical to the use of IM-PGA relationships in earthquake engineering and risk analysis in particular. Saying that the criticism is not made on the methodology and tools applied but on the philosophy (i.e. principles) of this methodology and the accuracy of the wished outcome."

**Response**: In our PGA-related analytical fragility database, the PGA parameter used actually is not the real instrumental records as used in traditional PGA-intensity relationship development method, which are collected from the same geographical range as macroseismic intensity records. Therefore, from this point of view, the regional dependence (here we mainly refer to site condition), which contributes to the scatter of traditional PGA-intensity relationship, is not a source of uncertainty in our relationship. Besides that, as aforementioned, the combination and synthetization of fragility from different building types makes the final PGA-intensity relationship become a very general one and not representative of any individual building type.

We'll further emphasis the limitation in potential application of our relationship.

Quotation of the general comment 6: "To the reviewer's opinion if the results of the present work i.e. the IM-PGA tables, should be used as recommended values for IM-PGA ranges in China, it should be clearly stated that this is just for preliminary evaluations and the scatter may be very important."

Response: Yes. Due to the scattering in originally collected fragility datasets and simplification in using median fragility to derive PGA-intensity relation, the potential application of the preliminary PGA-intensity relationship should be with caution. For engineering purposes, it is best to use regional relationships wherever available, as they are better calibrated for the areas in which they apply.

Quotation of the general comment: "In table7 (Recommendedintensity-PGA relationship in China (GB17742-2008/1980)) there is an obvious error in the suggested value for Intensity X."

Response: We appreciate your careful check very much. We'll rectify this from 0.1 to 1.0.

**References:**

Websites:

GEM Fragility Database: https://platform.openquake.org/vulnerability/list?type_of_assessment=1
GEM Vulnerability Database: https://www.ucl.ac.uk/epicentre/resources/gem-vulnerability-database
HAZUS User & Technical Manuals: https://www.fema.gov/hazus-mh-user-technical-manuals
PAGER Scientific Background: https://earthquake.usgs.gov/data/pager/background.php
PERPETUATE Project Report Summary: https://cordis.europa.eu/result/rcn/57689_en.html
SYNER-G Project Deliverables: http://www.vce.at/SYNER-G/files/dissemination/deliverables.html

Literature:

F.E.M.A., Multi-Hazard Loss Estimation Methodology, Earthquake Model, HAZUS-MH2. 1. 2012.
SYNER-G Project Report D3.1: Pitilakis, K., Systemic Seismic Vulnerability and Risk Analysis for Buildings, Lifeline Networks and Infrastructures Safety Gain. 2011.
SYNER-G Project Report D3.2: Pitilakis, K., Systemic Seismic Vulnerability and Risk Analysis for Buildings, Lifeline Networks and Infrastructures Safety Gain. 2011.
SYNER-G Project Report D3.12: Pitilakis, K., Systemic Seismic Vulnerability and Risk Analysis for Buildings, Lifeline Networks and Infrastructures Safety Gain. 2011.
Tang, B., et al., Evaluation of collapse resistance of RC frame structures for Chinese schools in seismic design categories B and C. Earthquake engineering and engineering vibration, 2011. 10(3): p. 369.
Yepes-Estrada, C., et al., The global earthquake model physical vulnerability database. Earthquake Spectra, 2016. 32(4): p. 2567--2585.

---

## Author Comment (AC3) · 15 Dec 2018

Quotation of the general comment: "They provide empirical fragilities for 4 building types. Also, they provide fragilities as functions of PGA using analytical methods. The authors say they "scrutinize 69 papers". However the 18 of them are "thesis". Is this acceptable?"

Response: Thank you for pointing this out. We'll replace "papers" by "publications". We didn't mean to hide the fact that 18 publications we scrutinized are thesis. Actually, some theses provided very good collection of empirical fragility data from historic earthquakes, e.g. Piao (2013, thesis), Ding (2016, thesis).

Quotation of the general comment: "According to the reviewers knowledge the macroseismic intensity is influenced by many factors (e.g.: population, economy level etc.). So earthquakes of the same magnitude may cause different macroseismic intensity levels in various cities. Moreover the time period is too long. The damage stage depends not only on the type of the building but on the kind of the construction and the construction type has changed during these years."

Response: Yes. We're deeply aware of the change of building fortification performance over the past 40-year time span. Therefore, the available building construction fortification level and age information are intergrated into the classification of buildings into masonry_A, masonry_B, RC_A, RC_B types, with reference to the criteria listed in Table 3 of the manuscript.

And we totally agree that uncertainty is prevailing in every single step both in empirical and analytical fragility analysis, as pointed out in Page 2 line 1-13 in the manuscript. In this regard, besides carefully classifying the originally collected fragility data and conducting box-plot analysis, we also provided the standard deviation of the overall fragility distribution for each intensity and PGA level in Appendix Figure A1-A4 and Table B1. Further beyond that, we also spare quite a few efforts in charactering the uncertainty transmission from the fragility curve to the PGA-intensity relationship based on the data in the Appendix.

To avoid the manuscript to appear to be too sparse and extensive, we didn't extend this uncertainty characterization process and only provide a reference uncertainty value of "0.3" in Page 9 line 9 for Eq. (5). However, for your further check, we put this methodology description in additionally uploaded file "Methodology of uncertainty transmission.pdf".
(This file is available at https://app.box.com/s/lwlqajpogxqlau72ravew4db7y47drtk)

Quotation of the general comment: "A list of the ground motions used for this study is required."

Response: We realized only the processed fragility data uploaded in the supplement is not enough. For your further check, here we also uploaded all the four original documents related to the preliminary processing of the collected fragility data: (available at https://app.box.com/s/lwlqajpogxqlau72ravew4db7y47drtk). Detailed explanation of each file and the building classification technique are given as follows:

(1) Filename "China Economic Loss and Vulnerability function review.xlsx":

This is our original records of fragility data extracted from the aforementioned scrutinized publications. Among those mentioned 87 publications, 71 have detailed fragilities given, the other 16 publications are either concept/method related or their data have been included in those 71 publications.

Sheet "site survey and statistic 36" includes empirical fragility data extracted from 36 publications without clear building fortification information, which will be estimated from available supplementary information;

Sheet "with fortification 16" includes empirical fragility data extracted from another 16 publications with clear building fortification information;

Sheet "analytical prediction 18" includes all the analytical fragility data extracted from 18 publications' tables or figures, among them some have clear building fortification information, others not;

Sheet "Ding Baorong 2016" is the fragility data collected by Ding (2016, doctoral thesis). Due to the high similarity in research purpose, we also digitalized these data and actually reiterated his/her work based the assumptions described in Ding (2016). This comparison will be explained later on.

(2) Filename "China Vulnerability records.xlsx":

Based on the results in Step (1), in this file the fragility data are further divided into different building groups: soil-wood, brick-wood, brick-concrete, analytical_masonry, RC, analytical_RC, industrial frame, stone-wood, Chuandou-timber, wood, stone and soil, as can be checked from each sheet with the same name. Here, 'brick-concrete' equals to masonry in our nomination. But for intensity-PGA relationship development, we have to focus only on masonry and RC, since analytical fragility data for other building types are not available.

(3) Filename "China Vulnerability analysis_plot.xlsx":

In this file, the exceeding probabilities of four damage limit states (LS1, LS2, LS3, LS4) are derived using Eq. (1) in the manuscript;

Given the main focus of this work as aforementioned and data abundance of each building type's fragility data, we finally focus on Sheets "brick-concrete", "analytical_masonry", "RC" and "analytical_RC" for further fragility curve derivation and PGA-intensity relationship exploration.

To fully use the fortification information given in each literature and make less assumption, we further add different fortification tags to empirical fragility data in Sheets "brick_concrete" and "RC", and to analytical fragility data in Sheets "analytical_masonry" and "analytical_RC";

The grouping criteria are as follows:

For empirical masonry fragility data in Sheet "brick-concrete": five tags specified in Column "Group of data" are used in differentiating the collected data: no fortification, unspecified fortification, low fortification, middle fortification and high fortification.

'no fortification' means there is no available information we can get from corresponding publication;

'unspecified fortification' means that in some publication, they mentioned the building is fortificated or unfortificated, but with no detailed fortification level information;

'low fortification' refers to buildings with VI level fortification as given in corresponding literature;

'middle fortification' refers to buildings with VII level fortification as given in corresponding literature;

'high fortification' refers to buildings with ≥VIII level fortification as given in corresponding literature.

Additionally, available building age information is used in grouping of fragility data extracted, like in Sheet "brick-concrete" from Hu & Sun (2010).

For analytical masonry fragility data in Sheet "analytical_masonry", in Column "Group of data":

'low-middle fortification' refers to buildings modelled with 0.05g~0.2g fortification as described in corresponding literature;

'high fortification' refers to buildings modelled with ≥0.3g fortification as described in corresponding literature.

For empirical RC fragility data in Sheet "steel-RC", the grouping criteria are similar to that in "brick-concrete", with slight difference in that given RC buildings are generally have better fortification performance than masonry, so in publications where building fortification information is not available, we mark it as "unspecified fortification" , as can be checked in Column "Group of data" as well.

For analytical RC fragility data in Sheet "analytical_steel", the grouping criteria are similar to that in "analytical_masonry", with slight difference that since RC buildings are generally have better fortification performance than masonry, so in publications where building fortification information is not available, we also mark it as "unspecified fortification".

Besides that, in Sheet "analytical_masonry", fragility data based both on PGA and SA are collected, but since for masonry building, only PGA related fragility data are available, so finally we only use PGA related analytical fragility data for RC buildings.

(4) Filename "China Vulnerability analysis_plot_result.xlsx":
This file is not so much different from the file in Step (3), only that we regroup the data with different fortification level assigned in Step (3) and plot the fragility distribution.

To achieve certain statistical significance of the fragility data analysis, in this step subjective judgement is necessary. Thus for brick-concrete or masonry buildings, we assign "masonry_A", "RC_A" building type to represent those with

unspericifed/low/middle fortification level tag in Step (3), and "masonry_B", "RC_B" to include those with high fortification level tag in Step (3), as also described in Page 5: line 18-23 in the manuscript.

The data uploaded in the online supplementary material (or the invalid personal link) is the classified fragility data extracted from this file.

Quotation of the general comment: "The link on page 5, line 27 does not work."

Response: The first author is quite sorry for the inconvenience caused due to the addition of the invalid personal link in Page 5: Line 27. It was meant to provide more access to the fragility data uploaded in the online supplementary material. We'll remove this link and directly recommend readers to download the online supplementary material, which is the same content as this invalid link refers to.

Quotation of the general comment: "Many references are not adequately written (e.g.: the journal is missing, or the pages are missing etc.)"

Response: Thank you very much for your careful check. The first author has no excuse for this kind of carelessness. After rechecking the 103 references, at least 10 of them are obviously incomplete. We'll definitely rectify that one by one.

**References:**

Ding, B.: Study on Related Quantitative Parameters of Seismic Intensity Scale, Thesis, Institute of Engineering Mechanics, China Earthquake Administration, Harbin, China, 195, 2016.
Piao, Y.: Study on Housing Seismic Vulnerability of Yunnan and Qinghai Province, Thesis, Institute of Engineering Mechanics, China Earthquake Administration, Harbin, China, 72, 2013.